# Elucidating the Design Space of FP4 training

## Abstract

The increasing computational demands of foundation models have spurred research into low-precision training, with 4-bit floating-point (`FP4`) formats emerging as a frontier for maximizing hardware throughput. While numerous techniques have been proposed to stabilize `FP4` training, they often present isolated solutions with varying, and not always clear, computational overheads. This paper aims to provide a unified view of the design space of `FP4` training. We introduce a comprehensive, quantisation gradient-based framework for microscaling quantization that allows for a theoretical analysis of the computational costs associated with different stabilization methods on both the forward and backward passes. Using a simulator built on this framework, we conduct an extensive empirical study across a wide range of machine learning tasks, including regression, image classification, diffusion models, and language models. By systematically evaluating thousands of combinations of techniques—such as novel gradient approximations, rounding strategies, and scaling methods, we identify which configurations offer the most favourable performance-to-overhead trade-off. We find that the techniques enabling the best trade-off involve carefully combining Hadamard transformations, tensor scaling and stochastic rounding. We further find that using `UE5M3` as a scaling factor potentially offers a good compromise between range and precision with manageable computational overhead.

## 1 Introduction

With the emergence of foundation models (Bommasani et al., 2021), the demand for computational resources has grown proportionally to the parameter count of these models, which often span from billions to trillions of parameters. Many of these models rely on the transformer architecture (Vaswani et al., 2017), which is ubiquitous across vision, text and video models (Khan et al., 2022). These models tend to be compute bound by two operations: the attention mechanism (Duman-Keles et al., 2023; Dolga et al., 2024) which tends to scale quadratically with sequence length, and the matmul operations from the weight matrices (Corp, 2025) which are quadratic with respect to size of the hidden dimensions.

In this paper, we conduct the first large-scale, systematic investigation of the `FP4` design space. Traditionally, machine learning training uses `FP32` format which serves as the baseline with the highest accuracy and lowest throughput. Historically, with each subsequent generation of hardware, the precision has been halved — starting from `FP16` (Micikevicius et al., 2018), `FP8` (Noune et al., 2022; Micikevicius et al., 2022; Fishman et al., 2025), and now `FP4` (Tseng et al., 2025; Chen et al., 2025; Castro et al., 2025; Hao et al., 2025; Chmiel et al., 2025; Wang et al., 2025; Yang et al., 2025; Su et al., 2025; Li et al., 2025; Cao et al., 2025). Halving the precision often allows a doubling of the throughput for matrix multiplication operations (Hao et al., 2025), hence with each iteration careful adjustments need to be made to the training procedure to account for the loss of numerical accuracy.

While these works introduce new techniques to stabilize `FP4` training for larger models, they all propose different methodologies with varying computational overhead that are empirically shown to work in isolation through simulations in `BFLOAT16`. However, a systematic evaluation of the performance-overhead trade-offs has been missing. As an example, Wang et al. (2025) introduces a quantile based pruning and gradient adjustment, both of which are shown to be useful, however both add an additional $\sim \mathcal{O}(n)$ time and memory overhead which cannot be done in low-precision and is non-fusable. It should be further noted that the simulation procedure in Wang et al. (2025) does not adequately quantise the scale, which their description implies is kept in high-precision – a detail

that can significantly impact training stability. Similarly, Tseng et al. (2025) proposes a block-wise Hadamard transformation, which induces a $\sim \mathcal{O}(n \log l)$ overhead. It should be noted that none of the aforementioned papers simulate $\mathtt{FP4}$ training fully, as Wang et al. (2025) only consider quantising weights and activations and not the gradient and Tseng et al. (2025) only quantises the gradient. Further, Cao et al. (2025); Li et al. (2025) proposes spectral decomposition techniques to handle outliers, which consequently introduces $\mathcal{O}(nk)$ time and $\mathcal{O}(k^2)$ memory, which in terms of hardware acceleration is also non-fusable. It is currently not clear whether these additional overheads are necessary in downstream implementations of low-precision matrix multiplication, as some evidence in Chmiel et al. (2025); Yang et al. (2025) suggests that something as simple as *Stochastic Rounding (SR)* is enough to stabilise $\mathtt{FP4}$ training. Here $l$ denotes block size, $n$ denotes number of elements in the tensor and $k$ the low-rank projection dimension in SVD.

The goal of this work is to develop a thorough understanding of the quantisation mechanism and how it affects the training procedure and illuminate which techniques offer a worthwhile trade-off in terms of additional overhead vs performance benefit. We summarise the contributions of this paper as follows:

1. We propose a quantisation gradient-based framework for $\mathtt{FP4}$ quantisation, which is used to derive the computational overhead of conceivably useful techniques (both novel and existing ones) on the forward and backward pass of a quantised linear layer.

2. We implement the framework as a simulator [1], running experiments of across various machine learning tasks to gain insight on which combination of techniques offer a reasonable overhead vs. performance benefit.

We first introduce our unified, gradient-based framework in Section 2, then use it to analyze the design space of scaling, rounding, and gradient approximation techniques in Sections 3, 4, and 5. We survey other relevant methods in Section 6, present our extensive empirical validation in Section 7, and conclude with our key findings.

## 2 A COMMON FRAMEWORK FOR $\mathtt{FP4}$ TRAINING

In this section, we detail what happens when we use microscaling formats to quantize a tensor for a linear layer forward pass. Consider a tensor $\mathbf{X} \in \mathbb{R}^{m \times n}$. We first define $\mathbf{X}$ represented in microscaling format Rouhani et al. (2023).

**Definition 1.** *A micro-scaled block is defined by a scalar $s \in \mathbb{R}$ and a vector $\mathbf{P} = [p_i]_{i=1}^{l}$ of $l$ elements. Each value $x_i$ can be recovered as*

$$x_i = s \cdot p_i.$$

*The parameter $l$ is a fixed constant known as the* block size*. Given a tensor $\mathbf{X} \in \mathbb{R}^{m \times n}$ and a block size of $l$, the* MX representation *of $\mathbf{X}$ consists of a collection of tuples*

$$\{(s_j, \mathbf{P}_j)\}_{j=1}^{(m \cdot n)//l},$$

*where each tuple corresponds to a block of $l$ elements in $\mathbf{X}$.*

Intuitively, microformat scaling represents partitions of a tensor with a common scale often used to normalise the partition, where the scaled elements $\mathbf{P}_j$ is quantized to a lower precision. We formally detail the quantisation procedure for one partition $\mathbf{X}_p \in \mathbb{R}^l$ below, represented by a transformation $f$:

$$f(\mathbf{X}_p) = \frac{1}{s_q} Q(s_q \cdot \mathbf{X}_p)$$

where the components are defined as follows:

**Outer Scaling Factor** $(s_q)$**:** This factor is a function of $\mathbf{X}_p$. First, an intermediate factor $s(\mathbf{X}_p)$ is computed:

$$s(\mathbf{X}_p) = \frac{\mathtt{FP4\ max}}{Z(\mathbf{X}_p)}$$

---

[1]Codebase: Google Drive Link

Here, $Z : \mathbb{R}^l \to \mathbb{R}$ is a scalar-valued function of the tensor $\mathbf{X}_p$ (e.g., the absolute maximum norm, `absmax`). This factor is then quantized:

$$s_q = q(s(\mathbf{X}_p))$$

**Quantization Function ($Q$):** $Q$ is an element-wise function that quantizes the elements of $\mathbf{X}_p$.

We can now introduce the gradient with respect to an element in $\mathbf{X}$:

**Proposition 1.** *Let $f(\mathbf{X}_p) = \frac{1}{s_q} Q(s_q \cdot \mathbf{X}_p)$ Then, the partial derivative of $f_{ij}$ with respect to $\mathbf{X}_{ij}$, that is the diagonal of the Jacobian matrix, is given by:*

$$\boxed{\frac{\partial f_{ij}}{\partial \mathbf{X}_{ij}} = Q'(s_q \mathbf{X}_{ij}) + \frac{\partial s}{\partial \mathbf{X}_{ij}} \left[ \frac{q'(s)}{s_q} \left( \mathbf{X}_{ij} Q'(s_q \mathbf{X}_{ij}) - \frac{1}{s_q} Q(s_q \mathbf{X}_{ij}) \right) \right]} \tag{1}$$

*See Appendix A.7 for derivations.*

In the context of a linear layer with weights $\mathbf{W} \in \mathbb{R}^{n \times m}$ and input data $\mathbf{X} \in \mathbb{R}^{b \times m}$, the output $\mathbf{Y} = f(\mathbf{X}) f(\mathbf{W})^\top$ would have the corresponding gradients:

$$\frac{\partial \mathcal{L}}{\partial \mathbf{X}} = \left( \frac{\partial \mathcal{L}}{\partial \mathbf{Y}} \cdot f(\mathbf{W}) \right) \odot \frac{\partial f(\mathbf{X})}{\partial \mathbf{X}}, \quad \frac{\partial \mathcal{L}}{\partial \mathbf{W}} = \left( \left( \frac{\partial \mathcal{L}}{\partial \mathbf{Y}} \right)^\top \cdot f(\mathbf{X}) \right) \odot \frac{\partial f(\mathbf{W})}{\partial \mathbf{W}}.$$

Here, $\mathcal{L}$ denotes the scalar loss, $\odot$ the elementwise product and $f(\cdot)$ is a differentiable transformation (e.g., MX decomposition or quantization-aware mapping) applied to the inputs and weights. In the next section, we detail different choices in terms of calculating and approximating $\frac{\partial \mathcal{L}}{\partial \mathbf{X}}, \frac{\partial \mathcal{L}}{\partial \mathbf{W}}$. We summarise the time and memory overhead of our proposed and existing techniques in Table 2.

## 3 TENSOR SCALING IN FP4 TRAINING

An alternative to applying block-wise scaling directly is to first normalize the entire tensor Blake et al. (2023); Micikevicius et al. (2022); Sun et al. (2019); Peng et al. (2023). The goal of this strategy is to improve the quantization of the scaling factors themselves. In this approach, a tensor-wise scaling factor $g$ is computed, used to normalize the tensor, and then multiplied back after the block-wise quantization. While the intent is for $g$ to cancel out, the non-linear nature of the scale quantization function $q(\cdot)$ results in a distinct final transformation.

Let $g = \max_p \{m_p\}$, $m_p = Z(\mathbf{X}_p)$ be the global scaling factor for a tensor $\mathbf{X}$, and let $\mathbf{U} = \mathbf{X}/g$ be the globally normalized tensor. The transformation $h(\mathbf{X})$ for an element $\mathbf{X}_{ij}$ within a block $p$ is defined as $h_{ij}(\mathbf{X}_p) = g \cdot f_{ij}(\mathbf{U}_p)$.

Here, $f(\mathbf{U}_p)$ is the block-wise quantization function from Section 2 applied to the normalized block $\mathbf{U}_p$. Its components are functions of $\mathbf{U}_p$:

1. **Ideal Scale:** $s'_p = \frac{\mathrm{FP4_{max}}}{Z(\mathbf{U}_p)} = g \cdot \frac{\mathrm{FP4_{max}}}{Z(\mathbf{X}_p)} = g \cdot s_p$

2. **Quantized Scale:** $s'_{q,p} = q(s'_p) = q(g \cdot s_p)$

Substituting these gives the full forward pass expression for an element:

$$\boxed{h_{ij}(\mathbf{X}) = \frac{g}{q(g \cdot s_p)} Q \left( q(g \cdot s_p) \cdot \frac{\mathbf{X}_{ij}}{g} \right)} \tag{2}$$

The gradient of this transformation accounts for both the block-wise dependencies and the global dependency on $g$.

**Corollary 1.** *Let $h(\mathbf{X})$ be the quantization with intermediate global normalization. For an element $\mathbf{X}_{ij}$ in block $p$, the partial derivative is:*

$$\boxed{\frac{\partial h_{ij}}{\partial \mathbf{X}_{ij}} = \frac{\partial f_{ij}}{\partial \mathbf{U}_{p,ij}} + \frac{\partial g}{\partial \mathbf{X}_{ij}} \left( f_{ij}(\mathbf{U}_p) - \mathbf{U}_{p,ij} \frac{\partial f_{ij}}{\partial \mathbf{U}_{p,ij}} \right)} \tag{3}$$

*where $\frac{\partial f_{ij}}{\partial \mathbf{U}_{p,ij}}$ is the full gradient from the first Theorem, evaluated on the normalized block $\mathbf{U}_p$ with its corresponding scales $(s'_p, s'_{q,p}, k'_p)$. See Appendix A.8 for derivations.*

It should be noted that $g$ in context of the `MXFP4`(FP4 format with `E8M0` scale) and `NVFP4` (FP4 format with `E4M3` scale) has overhead complexity $\mathcal{O}((m \cdot n)//l)$ as opposed to $\mathcal{O}(m \cdot n)$ when computing $\text{absmax}(\mathbf{X})$, since it suffices to search the scales rather than the entire tensor $\mathbf{X}$.

## 3.1 Rounding and Scaling Strategies

Recent work in low-precision training has highlighted that the rounding strategy for scaling factors can have a profound impact on model stability. For instance, Mishra et al. (2025) found that for `MXFP8` formats, rounding-to-positive-infinity improves signal propagation by reducing the number of saturated values, given the limited range of the scaling factor. As our work considers both `E4M3` (which has a limited range) and `E8M0` (which has a wider range), we evaluate both round-to-nearest (RTN) and round-to-positive-infinity in our experiments.

We note that the results in Chmiel et al. (2025) get `NVFP4` to converge without any issues with tensor scaling, as they mitigate any overflow by taking $\tilde{s}_p = \frac{s'_p}{\text{FP4}_{\max} \cdot \text{E4M3}_{\max} \cdot 0.5}$. This pushes down the effective range of $\tilde{s}_p \in [2/\text{E4M3}_{\max}, 2/\text{E4M3}_{\max} \cdot g)$. While not completely protected from overflow, it's a good rule of thumb to maximize the utilised range of `E4M3`. We use this technique when applying tensor scaling for `NVFP4`. We note that this heuristic can be extended to any scale format beyond `E4M3`, as it effectively rescales the scale factor to utilise its maximum range.

**Handling Zero-Valued Scales.** A critical edge case is the handling of zero-valued scaling factors, resulting in division by zero in the dequantisation. Chmiel et al. (2025) replaces any zero scale with one, which may induce further quantisation errors as small scales are set to 1. We propose rounding the zeros and underflows to the closest representable subnormal value in the target format and saturate overflows to the maximum representable number. We compare the efficacy of both approaches in our experiments.

**Rounding of the weight tensor** The impact of the rounding strategy has previously been demonstrated (Chmiel et al., 2025; Fitzgibbon & Felix, 2025) to have significant impact on the stability of LLM training in low-precision formats. The main observations for `FP4` formats is to use *round-to-nearest* (RTN) for the forward pass and *stochastic rounding* (SR) in the backwards pass (Chmiel et al., 2025; Yang et al., 2025), specifically on the activation and gradient tensors. We follow the quantisation procedure in Rouhani et al. (2023), which considers 6 quantisations for a forward and backward pass in a linear layer. We benchmark against the strategy proposed by Chmiel et al. (2025) and additionally consider SR on the activations in the forward pass as well.

**Rounding of the scales** We also experiment with stochastic rounding in the scaling factor as well. We motivate this design choice with the observation that `E8M0` has very large intervals between each number, leading to potential bias, which can mitigated more effectively at the scaling factor.

## 4 Differentiable Relaxations for Quantization

**Approximating $Q'(\mathbf{X})$ and $q'(x)$** Wang et al. (2025) take $\frac{\partial f_{ij}}{\partial \mathbf{W}_{ij}} \approx Q'(s \cdot \mathbf{W}_{ij})$. However since $Q$ is a quantisation function which is not differentiable, they approximate $Q(x) \approx \frac{\delta}{2} \cdot \left(1 + \text{sign}\left(\frac{2x}{\delta} - 1\right) \cdot \left|\frac{2x}{\delta} - 1\right|^{\frac{1}{w}}\right)$, with gradient $Q'(x) = \frac{1}{w} \cdot \left|\frac{2x}{\delta} - 1\right|^{\frac{1}{w} - 1}$. They propose $w = 5$ in their implementation. There are some potential flaws with the proposed parametrisation, as calculating power of fractionals tends to be computationally expensive and require $\mathcal{O}(w)$ cycles. This leads to the overall complexity of $\mathcal{O}(nmw \log_2(k))$, where $\log(k)$ comes from finding the interval on the `E2M1` grid to which $x$ belongs using binary search, with $k$ being the grid size. We thus propose an alternative differentiable relaxation of $Q(x)$.

**Linear Spline approximation** A linear spline is a continuous piecewise linear function defined over a set of sorted knots $t_0, \ldots, t_n$. These knots partition the domain into $n$ intervals $I_i = [t_i, t_{i+1})$. The function's continuity is ensured by having the linear segments connect at the knots.

The forward and backward passes evaluate the spline and its derivative. For an input $x \in [t_i, t_{i+1})$, the spline is a line segment, $S(x) = a_i(x - t_i) + b_i$ (**Forward pass**) with $S'(x) = a_i$ (**Backward pass**). Here, $b_i$ represents the value of the spline at knot $t_i$ (i.e., $S(t_i)$), and $a_i$ is the slope of the line segment over the interval $[t_i, t_{i+1}]$.

We illustrate our proposed differentiable quantization approximation and its corresponding gradient in Fig. 1. The function is shown in Fig. 1a, and its gradient is depicted in Fig. 1b. We found that applying the quantisation gradient in the backwards pass sometimes would mask out the gradient entirely, hence we propose clipping $Q'(x)$ from below to prevent multiplying the gradient with 0 (Figure 1c). The overall complexity overhead of the spline approximation is thus $\mathcal{O}(nm\log_2(k))$.

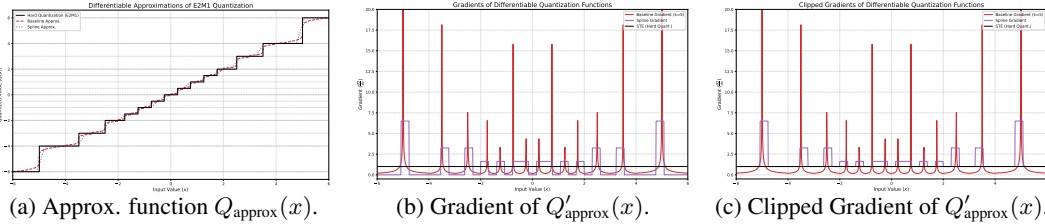

(a) Approx. function $Q_{\text{approx}}(x)$.    (b) Gradient of $Q'_{\text{approx}}(x)$.    (c) Clipped Gradient of $Q'_{\text{approx}}(x)$.

Figure 1: Approximations of $Q(x)$ and their corresponding gradients, assuming ties-to-even rounding. We refer to Wang et al. (2025) as the baseline.

Note that we need to save the unquantised matrix $\mathbf{X}$ for the backwards pass to evaluate $Q'(\mathbf{X})$, adding $\mathcal{O}(mn)$ memory overhead.

## 5 GRADIENT ADJUSTMENT FOR SCALING FACTOR QUANTIZATION

The gradient adjustment techniques used for weights and activations can also be applied to the quantization of the scaling factor $q(s)$. However, the relatively high dynamic range required for scaling factors introduces additional complexity. To find a decent trade-off between accuracy and complexity, we first analyze the regions where the quantization error is most significant. We measure this error using the relative deviation, defined as the ratio $s/s_q$. A value of this ratio far from 1 indicates a large quantization error.

Figure 2 illustrates the quantization functions for the E4M3 and E8M0 scale formats and their corresponding relative deviations. The quantization function itself is shown in Fig. 2a, while the error is plotted in Fig. 2b.

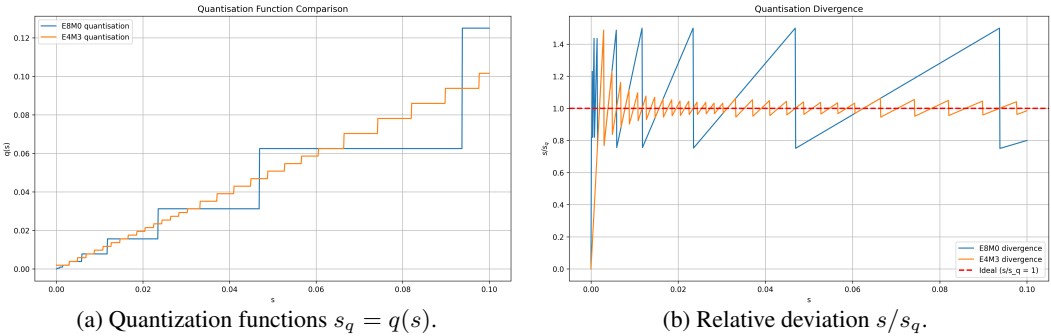

(a) Quantization functions $s_q = q(s)$.    (b) Relative deviation $s/s_q$.

Figure 2: Comparison of quantization for E4M3 and E8M0 scaling factors. Figure (a) shows the quantization step functions. Figure (b) shows the relative deviation, which is most pronounced for small values of the scaling factor $s$.

As illustrated in Fig. 2b, the largest relative deviation occurs for small-magnitude scaling factors, especially within the first few representable values of the E4M3 scale format. Based on this observation, we can choose to apply the gradient adjustment selectively, targeting only the range where the quantization error is highest when computing the $q'(s)$ term.

### 5.1 GRADIENT ADJUSTMENT OF ABSMAX: ADJUSTING FOR $Z(\mathbf{X}_p)$ AND $s'(\mathbf{X}_p)$

First, we establish the general relationship between the gradient of the scaling factor, $\frac{\partial s}{\partial \mathbf{X}}$, and the gradient of the normalization function, $\frac{\partial Z}{\partial \mathbf{X}}$.

**Proposition 2.** *Given the scaling factor $s(\mathbf{X}) = \frac{FP4\_max}{Z(\mathbf{X})}$, its gradient with respect to an element $\mathbf{X}_{ij}$ is given by:*

$$\frac{\partial s}{\partial \mathbf{X}_{ij}} = -\frac{FP4\_max}{Z(\mathbf{X})^2} \frac{\partial Z}{\partial \mathbf{X}_{ij}}$$

*See Appendix A.9 for derivations.*

The following corollaries provide the specific form of this gradient for two common choices of the normalization function $Z(\mathbf{X})$.

**Corollary 2.** *If the normalization function $Z(\mathbf{X})$ is* absmax, *$Z(\mathbf{X}) = \max_{k,l} |\mathbf{X}_{kl}|$, then the gradient of the scaling factor is non-zero only for the element with the maximum absolute value:*

$$\frac{\partial s}{\partial \mathbf{X}_{ij}} = -\frac{FP4\_max}{Z(\mathbf{X})^2} \left(\text{sign}(\mathbf{X}_{i^*j^*}) \cdot \delta_{ii^*} \delta_{jj^*}\right) \tag{4}$$

*where $(i^*, j^*)$ is the index of the maximum absolute value element and $\delta$ is the Kronecker delta. See Appendix A.10 for derivations.*

**Corollary 3.** *If the normalization function $Z(\mathbf{X})$ is the smooth LogSumExp approximation of the max function, $Z(\mathbf{X}) = \frac{1}{\beta} \log \left( \sum_{k,l} e^{\beta |\mathbf{X}_{kl}|} \right)$, the gradient of the scaling factor is a dense gradient given by:*

$$\frac{\partial s}{\partial \mathbf{X}_{ij}} = -\frac{FP4\_max}{Z(\mathbf{X})^2} \left(\text{softmax}(\beta|\mathbf{X}|)_{ij} \cdot \text{sign}(\mathbf{X}_{ij})\right) \tag{5}$$

*See Appendix A.11 for derivations.*

We consider four configurations for calculating the gradients with respect to the scaling factors, summarized in Table 1. Alongside the standard 'Absmax' and 'Softmax' approaches, we introduce a 'Hybrid' method. This approach uses the computationally efficient 'absmax' function in the forward pass but approximates its gradient with the dense 'softmax' derivative during the backward pass. This is intended to propagate gradient information to more elements without incurring the forward-pass cost of the LogSumExp operation.

Table 1: Gradient configurations for the block-wise scale $s(\mathbf{X})$ and global scale $g(\mathbf{X})$. The Straight-Through Estimator (STE) gradient is a heuristic approximation, as detailed in the text.

| Configuration | Scaling Function $Z(\mathbf{X})$ | Gradient $\frac{\partial s}{\partial \mathbf{X}_{ij}}$ | Global Scaling Function $g(\mathbf{X})$ | Gradient $\frac{\partial g}{\partial \mathbf{X}_{ij}}$ |
|---|---|---|---|---|
| **STE** | $\max_{k,l} |\mathbf{X}_{kl}|$ | $1$ | $\max_{k,l} |\mathbf{X}_{kl}|$ | $1$ |
| **Absmax** | $\max_{k,l} |\mathbf{X}_{kl}|$ | $-\frac{FP4\,max}{(Z(\mathbf{X}))^2} \left(\text{sign}(\mathbf{X}_{i^*j^*}) \cdot \delta_{ii^*} \delta_{jj^*}\right)$ | $\max_{k,l} |\mathbf{X}_{kl}|$ | $\text{sign}(\mathbf{X}_{i^*j^*}) \cdot \delta_{ii^*} \delta_{jj^*}$ |
| **Softmax** | $\frac{1}{\beta} \log \left( \sum_{k,l} e^{\beta |\mathbf{X}_{kl}|} \right)$ | $-\frac{FP4\,max}{(Z(\mathbf{X}))^2} \left(\text{softmax}(\beta|\mathbf{X}|)_{ij} \cdot \text{sign}(\mathbf{X}_{ij})\right)$ | $\frac{1}{\beta} \log \left( \sum_{k,l} e^{\beta |\mathbf{X}_{kl}|} \right)$ | $\text{softmax}(\beta|\mathbf{X}|)_{ij} \cdot \text{sign}(\mathbf{X}_{ij})$ |
| **Hybrid** | $\max_{k,l} |\mathbf{X}_{kl}|$ | $-\frac{FP4\,max}{(Z(\mathbf{X}))^2} \left(\text{softmax}(\beta|\mathbf{X}|)_{ij} \cdot \text{sign}(\mathbf{X}_{ij})\right)$ | $\max_{k,l} |\mathbf{X}_{kl}|$ | $\text{softmax}(\beta|\mathbf{X}|)_{ij} \cdot \text{sign}(\mathbf{X}_{ij})$ |

For any softmax-based configuration, we must either compute or save the softmax for the backwards pass, incurring additional time and memory complexity. For *Absmax*, it suffices to save the index of the maximum value. For the STE case of $\frac{\partial s}{\partial \mathbf{X}_{ij}}$, we are effectively setting the entire second term from Equation (1), $\left[ \frac{q'(s)}{s_q} \left( \mathbf{X}_{ij} Q'(s_q \mathbf{X}_{ij}) - \frac{1}{s_q} Q(s_q \mathbf{X}_{ij}) \right) \right]$, to be equal to 1. This provides a simple alternative to completely omit the $\mathcal{O}(3mn)$ extra computation of this extra gradient term, treating the complex scaling derivative as a direct pass-through. We have an `ignore` option for the $\frac{\partial g}{\partial \mathbf{X}_{ij}}$ term, which means setting the corresponding update term from Corollary 1 to zero: $\frac{\partial g}{\partial \mathbf{X}_{ij}} \left( f_{ij}(\mathbf{U}_p) - \mathbf{U}_{p,ij} \frac{\partial f_{ij}}{\partial \mathbf{U}_{p,ij}} \right) = 0$, skipping the extra $\mathcal{O}(4mn)$ work and saving memory.

## 6 OTHER TECHNIQUES

**Optimizer centric** Recently, Huang et al. (2025) proposed StableSPAM, which modifies the Adam optimiser by bounding the momentum term with a moving average statistic. This is motivated by the

Table 2: Summary of FP4 Training Techniques and Overheads. Here we assume each operation is applied to a tensor with $n$ elements, which can be partitioned to $n/l$ blocks with block size $l$.

| Technique | Compute Overhead | Additional Memory | Fuseability | Comment |
|---|---|---|---|---|
| Straight-Through Estimator (STE) | $\mathcal{O}(1)$ | None | Yes | |
| Baseline $Q'(\mathbf{X})$(Wang et al., 2025) | $\mathcal{O}(n \cdot w \log k)$ | $\mathcal{O}(n)$ | No | $w = 5$ |
| Spline $Q'(\mathbf{X})$ | $\mathcal{O}(n \log k)$ | $\mathcal{O}(n)$ | No | |
| Stochastic Rounding (Fitzgibbon & Felix, 2025) | $\mathcal{O}(n)$ | None | Yes | |
| Stochastic Rounding Scale | $\mathcal{O}(n//l)$ | None | Yes | |
| Global Tensor Scaling (Blake et al., 2023) | $\mathcal{O}(n)$ | $\mathcal{O}(1)$ | Yes | Rescale in full prec. |
| Global Scaling Gradient (Corollary 1) | $\mathcal{O}(3n)$ | $\mathcal{O}(3n)$ | No | Save ex. tensor |
| Differentiable Scale (Absmax) | $\mathcal{O}(4n)$ | $\mathcal{O}(3n)$ | No | |
| Differentiable Scale (Softmax) | $\mathcal{O}(4n)$ | $\mathcal{O}(4n)$ | No | Softmax backw. |
| Scale Gradient Adjustment | $\mathcal{O}(n)$ | $\mathcal{O}(n//l)$ | No | Only for Diff. Scale |
| Outlier concentration (Hadamard) (Tseng et al., 2025) | $\mathcal{O}(n \cdot \log l)$ | None | Yes | On-the-fly possible |
| StableSPAM Optimizer (Huang et al., 2025) | $\mathcal{O}(n)$ | $\mathcal{O}(1)$ | No | |
| Dynamic Loss Scaling (Micikevicius et al., 2018) | $\mathcal{O}(n)$ | $\mathcal{O}(1)$ | No | Mult. each tensor |
| SVD techniques (Li et al., 2025; Cao et al., 2025) | $\mathcal{O}(nk)$ | $\mathcal{O}(k^2)$ | No | |

observation that in low precision, the gradient norms tend to explode during training, meaning more careful normalisation of the momentum norm and bounding of large values is needed to stabilise training. While their optimiser is primarily tailored around LLMs, we explore the impact of combining StableSPAM with existing rounding and gradient adjustment based techniques for general purpose ML workloads.

**Loss scaling** We consider loss scaling as a technique to propagate signal when the range of the precision is very limited following Micikevicius et al. (2018). We implement the automated loss scaling technique, which adjusts the loss scaling scale dynamically during training.

**Outlier concentration** During quantisation, outliers in high-precision may induce quantisation error as they impact the scaling during quantisation. Recent work by Tseng et al. (2024; 2025) proposes applying $Q(\mathbf{HSX}_p)$ to concentrate outliers towards the median of the data. Here $\mathbf{HS}$ is the random Hadamard transform applied to each block $\mathbf{X}_p$ of size $l$ elements, inducing a $\mathcal{O}(\frac{mn}{l}l \cdot \log(l))$ compute overhead. It should be noted that this operation is fusable, and can be done on-the-fly with warp shuffle operations. We consider applying Hadamard transformation in both the forward and backward pass and only the backward pass, akin to Tseng et al. (2024; 2025); Castro et al. (2025).

**Spectral decomposition** In Li et al. (2025); Cao et al. (2025), they propose to use spectral decomposition techniques to alleviate the difficulty of quantising outliers in low-precision. This is done by decomposing the tensor into a low-rank representation using *singular value decomposition* (SVD), where the low-rank components are then quantised instead. As this is non-fusable, and has prohibitive time complexity overhead $\mathcal{O}(mnk)$ (with $k$ referring to a chosen lower rank), we do not consider it for our simulations as the Hadamard transformation offers a more seamless alternative in the pre-training setting.

## 7 EXPERIMENTS

**Experimental design and selection strategy** We consider the search space in Appendix Table 10, which totals to over 20,000 different parameter combinations, an infeasible search space for larger models. Consequently, our strategy is to do larger sweeps for smaller models that are faster to train and using the results to derive insights and prune the search space for larger models. We run the experiments in the order described in Appendix Table 11.

**Performance–efficiency score** We define an efficiency score $S(c) = \frac{G(c)}{1+\Omega(c)}$, for a configuration $c$, that balances relative performance gain $G(c) = (M_{\text{ref}} - M_c)/M_{\text{ref}}$ against a complexity penalty $\Omega(c) = \sum_{t \in \mathcal{T}_c} w_t$. Here, $\mathcal{T}_c$ is the set of non-standard techniques used, $w_t$ their overhead points, and the $+1$ ensures a well-defined score for baseline configurations. Scores are split by positive/negative gain per format, guiding pruning toward configurations that maximize performance with minimal added complexity (see Appendix A.6 for more details). We consider validation loss for $M$ when we calculate the score.

## 7.1 RESULTS

Loss curves of ImageNet-100, Gaussian regression, U-net large (big_diffusion) and Llama 60M, 350M and 1B are in Figure 3. For detailed results of each dataset we refer to Appendices A.2 and A.5. Based on our learnings from experiments, we present three guiding principles when training in FP4.

Figure 3: Training and validation performance curves for selected models and datasets.

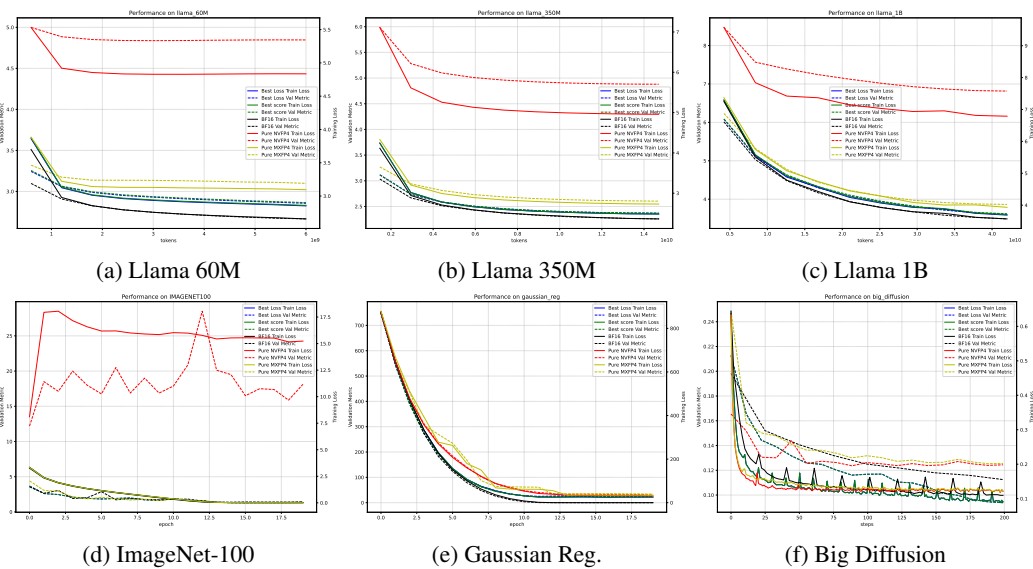

(a) Llama 60M    (b) Llama 350M    (c) Llama 1B

(d) ImageNet-100    (e) Gaussian Reg.    (f) Big Diffusion

**Principle 1: Gradient Stability Outweighs Unbiasedness** Across all our experiments (Appendix A.2) we found that none of the proposed gradient adjustment had any significant positive effect on training stability compared to STE and consequently we were unable to match the findings in Wang et al. (2025). As a possible explanation for this, consider the `absmax` gradient (Corollary 2), which is a single non-zero entry per block. This mathematically enforces a sparse, high-variance update signal that may introduce high-variance, impacting momentum based optimisers such as Adam or StableSPAM. We also observed this when experimenting with $Q'(\mathbf{X})$, that adding this gradient without lower and upper bound clipping of the relaxation (see Figure 1c) ended up masking out the downstream signal. Consequently using STE, which offers a dense and stable update and led to more stable training in the low-precision context.

**Principle 2: Scale Representation is the Primary Bottleneck** We find throughout our experiments and ablations studies that the range of the scaling factor has a profound effect on training stability, especially demonstrated in larger language models and the ImageNet-100 runs in Appendix Tables 3 and 4. As many of our results contradicted findings in Chmiel et al. (2025), we ran ablation studies in Appendix A.3 investigating the additional impact of `SmoothSwiGLU`[2] (Fishman et al., 2025) on language models and ablating the range of `E4M3` scaling, by replacing it with `E8M3` in Appendix A.4. Our findings suggest that `E4M3`, despite applying tensor scaling, did not converge due to its range limitation. We speculate that a potential sweet spot exists between `E8M0` and `E4M3`. We then experiment with `UE5M3` in Appendix A.5, a format that has increased range and additional precision and find that it indeed consistently outperforms `E8M0` on language modelling. The caveat however is that it requires tensor scaling and SR in the backwards pass to achieve this performance. We further find throughout our experiments that NaN-handling also depends on scale format and that although it doesn't have a big impact, it matters.

**Principle 3: The Performance-Overhead Frontier is Sparse** From our sweeps over thousands of configurations, we find that only a handful of techniques such as Hadamard transforms, tensor scal-

---

[2]This operation adds an approximate $\sim \mathcal{O}(n)$ non-fusable overhead due to additional `absmax` normalisation.

ing, stochastic rounding and optimiser choice provide a consistent, positive return on their computational overhead. We illustrate this in Pareto-frontier plots in Figure 4 for each dataset and Appendix Figure 10 for the `UE5M3` experiments. We overall observe that less complex configurations achieve better scores, and adding complexity yields diminishing returns. One can achieve lower loss, but often at a significant cost with respect to increased overhead, as observed in classification tasks.

Figure 4: Pareto-frontier plots for each dataset, $\Omega(c)$ on the x-axis and $S(c)$ on the y-axis. $S(c) = 0$ implies the configuration $c$ matches `BFLOAT16` performance.

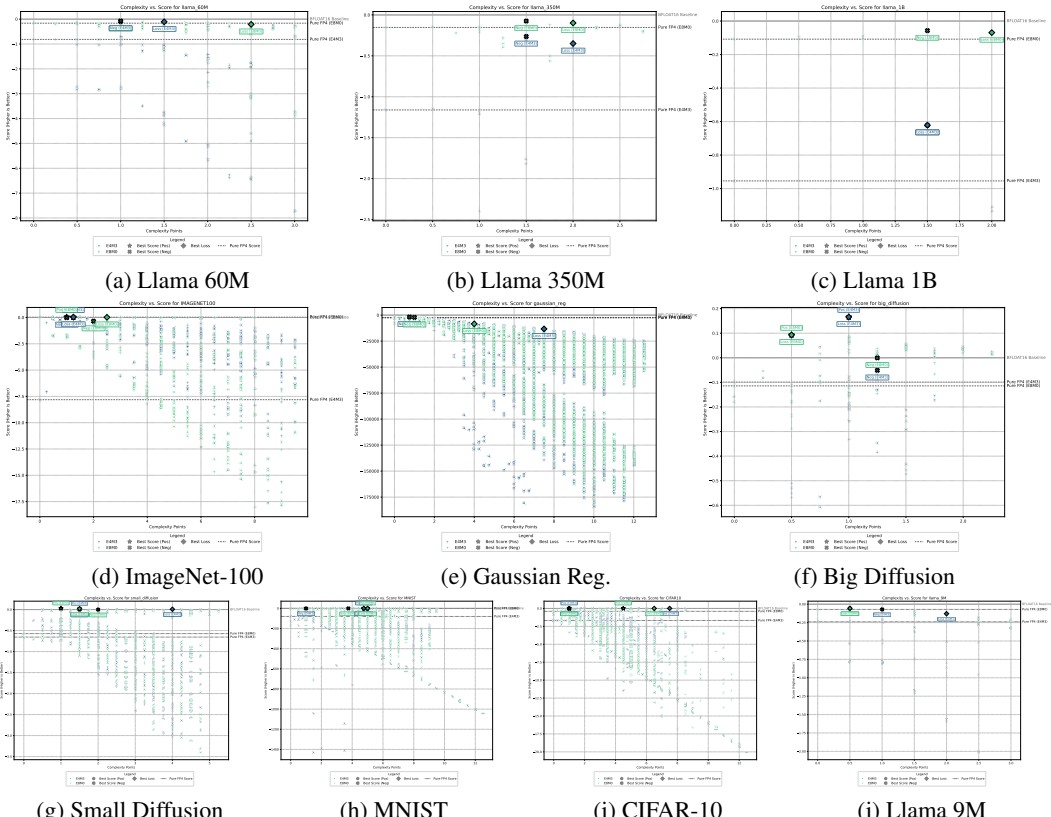

|                    |                  |                    |
|--------------------|------------------|--------------------|
| (a) Llama 60M      | (b) Llama 350M   | (c) Llama 1B       |
| (d) ImageNet-100   | (e) Gaussian Reg.| (f) Big Diffusion  |
| (g) Small Diffusion| (h) MNIST        | (i) CIFAR-10       |

(j) Llama 9M

**Conclusion and further work** We propose a novel framework for deriving the exact gradient updates for a linear layer under micro-scaling quantisation. While differentiable `absmax` gradients and quantisation gradients provided a benefit on smaller classification tasks, we found they offered no improvement or were detrimental for larger diffusion and language models, suggesting they can often be omitted to reduce overhead without sacrificing performance in these domains. Stochastic rounding of the scale showed little success beyond small models. We further find that the range of `NVFP4` hampers its performance on language models, and that the format might require additional overhead inducing adjustments beyond what is presented in Chmiel et al. (2025) for language models up to 1B. We find that `UE5M3` scale yields better results than `MXFP4`, offering a compromise between range and precision, however requiring tensor scaling and SR to work for LLM training, introducing, albeit manageable, overhead. For further research on hardware supporting FP4 training, we'd recommend starting out with `MXFP4` and implementing fusable operations such as Hadamard transformation, SR, Tensor scaling, being mindful of NaN-handling, carefully selecting the optimiser and finally exploring different scaling formats such as `UE5M3`. Finally, our work highlights that FP4 training dynamics may not be consistent across model scales, and leave this critical direction for further research.

**Reproducibility statement**   We provide the entire codebase in this Google Drive Link. One can use `uv` applied to the `.pytoml` file to reproduce the environment. Our code will run on commonly available hardware compatible with `PyTorch`. We've only used open-source datasets and included the download scripts in the codebase. We've ensured to fix all the seeds for all our runs (defined in the code), to hopefully provide total consistent and reproducible runs.

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
