# A  SUPPLEMENTARY APPENDIX

## A.1  LLM USE DISCLOSURE

LLMs were used in writing this paper. LLMs were used to:

1. Polish the writing, wording and condensing text

2. Parse tedious mathematical derivations into latex

3. Parse tedious figures and tables into latex

4. Helping write some of the code

## A.2  EXPERIMENTAL RESULTS

We provide detailed experimental results in this section.

**Linear regression** We first consider a linear regression task with synthetically generated data $y = X \cdot w_{\text{true}}$ for $X \in \mathbb{R}^{100000 \times 1024}$, $w_{\text{true}} \in \mathbb{R}^{1024 \times 1}$ with $X_{ij}, (w_{\text{true}})_i \overset{\text{iid}}{\sim} \mathcal{N}(0, 1)$. We present our results in Table 3. We find that the StableSPAM optimiser finds a perfect solution. We find that stochastic rounding applied to an E4M3 scale yielded the best trade-off in results. Additional gradients resulting from the absmax normalisation were not found to be helpful. Overall the StableSPAM optimiser remains the superior choice for regression.

Table 3: Experimental results

| Dataset | Source | Val loss | Train loss | Scale | Block size | Max grad. | Quant. grad | Hadamard | Scale grad | SR | Optimiser | Loss scaling | Round mode | Tensor scaling | Tensor grad | Complexity points | Score | NaN mode |
|---|---|---|---|---|---|---|---|---|---|---|---|---|---|---|---|---|---|---|
| IMAGENET100 | Baseline | 1.383 | 0.014 | N/A | N/A | N/A | N/A | N/A | N/A | N/A | Adam | False | N/A | N/A | N/A | N/A | N/A | N/A |
| IMAGENET100 | Baseline | 1.750 | 0.078 | N/A | N/A | N/A | N/A | N/A | N/A | N/A | StableSPAM | False | N/A | N/A | N/A | N/A | N/A | N/A |
| IMAGENET100 | Best Score (Neg) | 1.391 | 0.018 | E4M3 | 16 | STE | STE | N/A | STE | None_exact | Adam | True | TowardPositive | True | ignore | 1.000 | -0.006 | nearest_subnormal |
| IMAGENET100 | Best Score (Neg) | 1.625 | 0.087 | E8M0 | 32 | STE | STE | N/A | STE | all_activation_exact | StableSPAM | True | TiesToEven | True | ignore | 2.000 | -0.350 | nearest_subnormal |
| IMAGENET100 | Best Score (Pos) | 1.320 | 0.015 | E4M3 | 16 | STE | STE | N/A | STE | IntelFP4_exact | Adam | True | Stochastic | False | N/A | 1.250 | 0.036 | nearest_subnormal |
| IMAGENET100 | Best Score (Pos) | 1.312 | 0.014 | E8M0 | 32 | STE | STE | N/A | STE | None_exact | Adam | True | TowardPositive | True | ignore | 1.000 | 0.051 | nearest_subnormal |
| IMAGENET100 | Best loss MXFP4 | 1.312 | 0.014 | E8M0 | 32 | STE | STE | N/A | spline | None_exact | Adam | True | TowardPositive | True | ignore | 2.500 | 0.020 | nearest_subnormal |
| IMAGENET100 | Best loss NVFP4 | 1.320 | 0.015 | E4M3 | 16 | STE | STE | N/A | STE | IntelFP4_exact | Adam | True | Stochastic | False | N/A | 1.250 | 0.036 | to_one |
| IMAGENET100 | Pure FP4 | 12.188 | 8.112 | E4M3 | 16 | STE | STE | N/A | STE | None_exact | Adam | False | TiesToEven | False | N/A | 0.000 | -7.814 | nearest_subnormal |
| IMAGENET100 | Pure FP4 | 1.344 | 0.014 | E8M0 | 32 | STE | STE | N/A | STE | None_exact | Adam | False | TiesToEven | False | N/A | 0.000 | 0.028 | nearest_subnormal |
| big_diffusion | Baseline | 0.135 | 0.128 | N/A | N/A | N/A | N/A | N/A | N/A | N/A | Adam | False | N/A | N/A | N/A | N/A | N/A | N/A |
| big_diffusion | Baseline | 0.113 | 0.110 | N/A | N/A | N/A | N/A | N/A | N/A | N/A | StableSPAM | False | N/A | N/A | N/A | N/A | N/A | N/A |
| big_diffusion | Best Score (Neg) | 0.117 | 0.113 | E4M3 | 16 | STE | STE | N/A | STE | IntelFP4_exact | Adam | True | Stochastic | False | N/A | 1.250 | -0.051 | nearest_subnormal |
| big_diffusion | Best Score (Neg) | 0.113 | 0.108 | E8M0 | 32 | STE | STE | N/A | STE | all_activation_exact | StableSPAM | False | Stochastic | False | N/A | 1.250 | -0.000 | nearest_subnormal |
| big_diffusion | Best Score (Pos) | 0.094 | 0.088 | E4M3 | 16 | STE | STE | N/A | STE | None_exact | StableSPAM | False | TiesToEven | True | ignore | 1.000 | 0.166 | nearest_subnormal |
| big_diffusion | Best Score (Pos) | 0.102 | 0.095 | E8M0 | 32 | STE | STE | N/A | STE | None_exact | StableSPAM | False | TiesToEven | False | N/A | 0.500 | 0.093 | nearest_subnormal |
| big_diffusion | Best loss MXFP4 | 0.102 | 0.095 | E8M0 | 32 | STE | STE | N/A | STE | None_exact | StableSPAM | False | TiesToEven | False | N/A | 0.500 | 0.093 | nearest_subnormal |
| big_diffusion | Best loss NVFP4 | 0.094 | 0.088 | E4M3 | 16 | STE | STE | N/A | STE | None_exact | StableSPAM | False | TiesToEven | True | ignore | 1.000 | 0.166 | nearest_subnormal |
| big_diffusion | Pure FP4 | 0.124 | 0.117 | E4M3 | 16 | STE | STE | N/A | STE | None_exact | Adam | False | TowardPositive | False | N/A | 0.000 | -0.099 | nearest_subnormal |
| big_diffusion | Pure FP4 | 0.125 | 0.118 | E8M0 | 32 | STE | STE | N/A | STE | None_exact | Adam | False | TowardPositive | False | N/A | 0.000 | -0.114 | nearest_subnormal |
| gaussian_reg | Baseline | 25.250 | 25.224 | N/A | N/A | N/A | N/A | N/A | N/A | N/A | Adam | False | N/A | N/A | N/A | N/A | N/A | N/A |
| gaussian_reg | Baseline | 0.013 | 0.013 | N/A | N/A | N/A | N/A | N/A | N/A | N/A | StableSPAM | False | N/A | N/A | N/A | N/A | N/A | N/A |
| gaussian_reg | Best Score (Neg) | 23.500 | 24.875 | E4M3 | 16 | STE | STE | N/A | STE | None_exact | StableSPAM | False | Stochastic | False | N/A | 0.750 | -1815.151 | nearest_subnormal |
| gaussian_reg | Best Score (Neg) | 27.875 | 27.928 | E8M0 | 32 | STE | STE | N/A | STE | None_exact | StableSPAM | False | TowardPositive | True | ignore | 1.000 | -2153.264 | nearest_subnormal |
| gaussian_reg | Best loss MXFP4 | 27.250 | 30.474 | E8M0 | 32 | STE | spline | backward_exact | STE | None_exact | StableSPAM | False | TowardPositive | True | ignore | 4.000 | -8419.849 | nearest_subnormal |
| gaussian_reg | Best loss NVFP4 | 22.875 | 24.467 | E4M3 | 16 | STE | spline | backward_exact | STE | None_exact | StableSPAM | True | TiesToEven | True | absmax | 7.500 | -13251.368 | nearest_subnormal |
| gaussian_reg | Pure FP4 | 30.125 | 30.947 | E4M3 | 16 | STE | STE | N/A | STE | None_exact | Adam | False | TiesToEven | False | N/A | 0.000 | -2327.151 | nearest_subnormal |
| gaussian_reg | Pure FP4 | 34.000 | 34.303 | E8M0 | 32 | STE | STE | N/A | STE | None_exact | Adam | False | TiesToEven | False | N/A | 0.000 | -2626.623 | nearest_subnormal |

**Image classification** We find in Table 3 and Appendix Table 5, that any max relaxation had no effect on performance. Here we find that in some cases using absmax in the tensor scaling gradient tends to help. We generally find that stochastic rounding, combined with tensor scaling and loss scaling leads to the most effective improvements. Here Adam seems to work better overall. We observe that NVFP4 does not work out of the box while MXFP4 does.

**Diffusion** We find in Table 3 and and Appendix Table 5 that any application of absmax gradients does not have any positive effect. We find that it suffices to use loss and tensor scaling combined with the StableSPAM optimiser to achieve a good performance.

**LLM** We present our results in Table 4 and Figure 3. We are unable to reproduce the findings of Chmiel et al. (2025), where we contrastingly find MXFP4 to outperform NVFP4 for LLM training. We note that Fishman et al. (2025) additionally uses SmoothSwiGLU Fishman et al. (2025) in their experiments which induces a non-fusable $\sim \mathcal{O}(n)$ overhead as it requires the absmax along one dimension of the tensor, which we have omitted in our main experiments. We include this in further experiments in Appendix A.3 and find that it marginally improves performance, but still fails for the 1B model.

In contrast to Tseng et al. (2025); Castro et al. (2025), we did not find that the combination of Hadamard transformation and SR yielded a significantly better result for MXFP4, suggesting that SR can possibly be omitted to reduce overhead. We do not find that the relaxation of quantisation gradients proposed in Zhou et al. (2025) had any impact on stabilizing the training of LLMs.

Table 4: LLM results

| Dataset | Source | Val loss | Train loss | Scale | Block size | Max grad. | Quant. grad | Hadamard | Scale grad | SR | Optimiser | Loss scaling | Round mode | Tensor scaling | Tensor grad | Complexity points | Score | NaN mode |
|---|---|---|---|---|---|---|---|---|---|---|---|---|---|---|---|---|---|---|
| llama_1B | Baseline | 3.578 | 3.682 | N/A | N/A | N/A | N/A | N/A | N/A | N/A | Adam | False | N/A | N/A | N/A | N/A | N/A | N/A |
| llama_1B | Baseline | 3.487 | 3.569 | N/A | N/A | N/A | N/A | N/A | N/A | N/A | StableSPAM | False | N/A | N/A | N/A | N/A | N/A | nearest_subnormal |
| llama_1B | Best Score (Neg) | 4.933 | 4.957 | E4M3 | 16 | STE | STE | N/A | STE | None_exact | StableSPAM | False | TiesToEven | True | ignore | 1.500 | -0.622 | nearest_subnormal |
| llama_1B | Best Score (Neg) | 3.620 | 3.701 | E8M0 | 32 | STE | STE | all_exact | STE | None_exact | StableSPAM | False | TiesToEven | False | N/A | 1.500 | -0.057 | nearest_subnormal |
| llama_1B | Best loss MXFP4 | 3.608 | 3.688 | E8M0 | 32 | STE | STE | all_exact | STE | IntelFP4_exact | StableSPAM | False | TiesToEven | False | N/A | 2.000 | -0.069 | nearest_subnormal |
| llama_1B | Best loss NVFP4 | 4.933 | 4.957 | E4M3 | 16 | STE | STE | N/A | STE | None_exact | StableSPAM | True | TiesToEven | True | ignore | 1.500 | -0.622 | nearest_subnormal |
| llama_1B | Pure FP4 | 6.815 | 6.789 | E4M3 | 16 | STE | STE | N/A | STE | None_exact | Adam | False | TiesToEven | False | N/A | 0.000 | -0.954 | nearest_subnormal |
| llama_1B | Pure FP4 | 3.864 | 3.932 | E8M0 | 32 | STE | STE | N/A | STE | None_exact | Adam | False | TiesToEven | False | N/A | 0.000 | -0.108 | nearest_subnormal |
| llama_350M | Baseline | 2.269 | 2.375 | N/A | N/A | N/A | N/A | N/A | N/A | N/A | Adam | False | N/A | N/A | N/A | N/A | N/A | N/A |
| llama_350M | Baseline | 2.258 | 2.363 | N/A | N/A | N/A | N/A | N/A | N/A | N/A | StableSPAM | False | N/A | N/A | N/A | N/A | N/A | nearest_subnormal |
| llama_350M | Best Score (Neg) | 2.655 | 2.783 | E4M3 | 16 | STE | STE | N/A | STE | IntelFP4_exact | StableSPAM | False | TiesToEven | True | ignore | 1.500 | -0.264 | nearest_subnormal |
| llama_350M | Best Score (Neg) | 2.371 | 2.485 | E8M0 | 32 | STE | STE | all_exact | STE | None_exact | StableSPAM | False | TiesToEven | False | N/A | 1.500 | -0.075 | nearest_subnormal |
| llama_350M | Best loss MXFP4 | 2.369 | 2.483 | E8M0 | 32 | STE | STE | all_exact | STE | None_exact | StableSPAM | False | TiesToEven | True | ignore | 2.000 | -0.098 | nearest_subnormal |
| llama_350M | Best loss NVFP4 | 2.653 | 2.781 | E4M3 | 16 | STE | STE | N/A | STE | IntelFP4_exact | StableSPAM | True | TiesToEven | True | ignore | 2.000 | -0.350 | nearest_subnormal |
| llama_350M | Pure FP4 | 4.880 | 4.958 | E4M3 | 16 | STE | STE | N/A | STE | None_exact | Adam | False | TiesToEven | False | N/A | 0.000 | -1.161 | nearest_subnormal |
| llama_350M | Pure FP4 | 2.603 | 2.731 | E8M0 | 32 | STE | STE | N/A | STE | None_exact | Adam | False | TiesToEven | False | N/A | 0.000 | -0.153 | nearest_subnormal |
| llama_60M | Baseline | 2.665 | 2.657 | N/A | N/A | N/A | N/A | N/A | N/A | N/A | Adam | False | N/A | N/A | N/A | N/A | N/A | N/A |
| llama_60M | Baseline | 2.983 | 3.028 | N/A | N/A | N/A | N/A | N/A | N/A | N/A | StableSPAM | False | N/A | N/A | N/A | N/A | N/A | nearest_subnormal |
| llama_60M | Best Score (Neg) | 2.864 | 2.860 | E4M3 | 16 | STE | STE | N/A | STE | IntelFP4_exact | Adam | False | TiesToEven | True | ignore | 1.000 | -0.074 | nearest_subnormal |
| llama_60M | Best Score (Neg) | 2.917 | 2.908 | E8M0 | 32 | STE | STE | all_exact | STE | None_exact | Adam | False | TiesToEven | False | N/A | 1.000 | -0.094 | nearest_subnormal |
| llama_60M | Best loss MXFP4 | 2.889 | 2.880 | E8M0 | 32 | STE | STE | all_exact | STE | None_exact | StableSPAM | True | TiesToEven | True | ignore | 2.500 | -0.210 | to_one |
| llama_60M | Best loss NVFP4 | 2.856 | 2.852 | E4M3 | 16 | STE | STE | N/A | STE | IntelFP4_exact | StableSPAM | False | TiesToEven | True | ignore | 1.500 | -0.107 | nearest_subnormal |
| llama_60M | Pure FP4 | 4.838 | 4.829 | E4M3 | 16 | STE | STE | N/A | STE | None_exact | Adam | False | TiesToEven | False | N/A | 0.000 | -0.815 | nearest_subnormal |
| llama_60M | Pure FP4 | 3.099 | 3.096 | E8M0 | 32 | STE | STE | N/A | STE | None_exact | Adam | False | TiesToEven | False | N/A | 0.000 | -0.163 | nearest_subnormal |

**Exploring `UE5M3` scale format**   We find in ablation studies (see Appendix A.4) for `E4M3`, that the limiting factor during LLM training (with tensor scaling only) is the range of the exponent. We explore whether an alternative format like `UE5M3` can achieve better performance than `MXFP4` in Appendix A.5. Our results suggest that `UE5M3` offers a good compromise, with improved performance compared to `E8M0` scale on language modelling tasks. A caveat however is that `UE5M3` needs tensor scaling and SR in the backwards pass to stabilise, and exhibits instability in its pure form, unlike `MXFP4`. There is thus a computational overhead needed for the increased precision. We note that the best nan-handling strategy changes to "to_one".

**Additional dataset results**   We present the additional results for MNIST, CIFAR10, Llama 9M and Small U-net (CIFAR 10) in Figure 5 and Table 5.

Figure 5: Training and validation performance curves for other datasets.

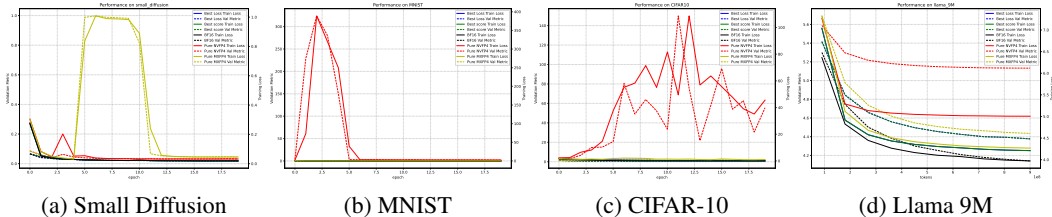

(a) Small Diffusion  (b) MNIST  (c) CIFAR-10  (d) Llama 9M

Table 5: Additional experimental results

| Dataset | Source | Val loss | Train loss | Scale | Block size | Max grad. | Quant. grad | Hadamard | Scale grad | SR | Optimiser | Loss scaling | Round mode | Tensor scaling | Tensor grad | Complexity points | Score | NaN mode |
|---|---|---|---|---|---|---|---|---|---|---|---|---|---|---|---|---|---|---|
| CIFAR10 | Baseline | 0.875 | 0.003 | N/A | N/A | N/A | N/A | N/A | N/A | N/A | Adam | False | N/A | N/A | N/A | N/A | N/A | N/A |
| CIFAR10 | Baseline | 0.895 | 0.027 | N/A | N/A | N/A | N/A | N/A | N/A | N/A | StableSPAM | False | N/A | N/A | N/A | N/A | N/A | N/A |
| CIFAR10 | Best Score (Neg) | 0.883 | 0.005 | E4M3 | 16 | STE | STE | N/A | STE | None_exact | Adam | True | TowardPositive | True | ignore | 1.000 | -0.009 | nearest_subnormal |
| CIFAR10 | Best Score (Neg) | 0.883 | 0.003 | E8M0 | 32 | STE | STE | N/A | STE | IntelFP4_exact | Adam | False | TiesToEven | True | ignore | 1.000 | -0.009 | nearest_subnormal |
| CIFAR10 | Best Score (Pos) | 0.867 | 0.003 | E4M3 | 16 | STE | STE | N/A | STE | all_activation_exact | Adam | True | TiesToEven | False | N/A | 1.000 | 0.009 | nearest_subnormal |
| CIFAR10 | Best loss MXFP4 | 0.855 | 0.040 | E8M0 | 32 | STE | STE | N/A | STE | all_activation_exact | Adam | True | TiesToEven | True | absmax | 4.500 | 0.005 | nearest_subnormal |
| CIFAR10 | Best loss MXFP4 | 0.855 | 0.040 | E8M0 | 32 | STE | spline | N/A | STE | all_activation_exact | Adam | True | TiesToEven | True | absmax | 6.500 | 0.003 | nearest_subnormal |
| CIFAR10 | Best loss NVFP4 | 0.836 | 0.037 | E4M3 | 16 | STE | spline | N/A | STE | None_exact | Adam | True | TowardPositive | True | absmax | 7.500 | 0.006 | nearest_subnormal |
| CIFAR10 | Pure FP4 | 2.344 | 2.354 | E4M3 | 16 | STE | STE | N/A | STE | None_exact | Adam | False | TowardPositive | False | N/A | 0.000 | -1.679 | nearest_subnormal |
| CIFAR10 | Pure FP4 | 1.227 | 0.911 | E8M0 | 32 | STE | STE | N/A | STE | None_exact | Adam | False | TiesToEven | False | N/A | 0.000 | -0.402 | nearest_subnormal |
| MNIST | Baseline | 0.027 | 0.016 | N/A | N/A | N/A | N/A | N/A | N/A | N/A | Adam | False | N/A | N/A | N/A | N/A | N/A | N/A |
| MNIST | Baseline | 0.028 | 0.004 | N/A | N/A | N/A | N/A | N/A | N/A | N/A | StableSPAM | False | N/A | N/A | N/A | N/A | N/A | N/A |
| MNIST | Best Score (Neg) | 0.027 | 0.008 | E4M3 | 16 | STE | STE | N/A | STE | None_exact | StableSPAM | True | TowardPositive | False | N/A | 1.000 | -0.005 | nearest_subnormal |
| MNIST | Best Score (Neg) | 0.027 | 0.007 | E8M0 | 32 | STE | spline | N/A | STE | None_exact | StableSPAM | True | Stochastic | True | ignore | 3.750 | -0.051 | nearest_subnormal |
| MNIST | Best Score (Pos) | 0.021 | 0.009 | E4M3 | 16 | STE | STE | N/A | STE | None_exact | StableSPAM | True | Stochastic | True | absmax | 4.750 | 0.050 | nearest_subnormal |
| MNIST | Best Score (Pos) | 0.021 | 0.006 | E8M0 | 32 | STE | STE | N/A | STE | IntelFP4_exact | StableSPAM | True | TiesToEven | True | absmax | 5.000 | 0.043 | nearest_subnormal |
| MNIST | Best loss MXFP4 | 0.021 | 0.006 | E8M0 | 32 | STE | STE | N/A | STE | IntelFP4_exact | StableSPAM | True | TiesToEven | True | absmax | 5.000 | 0.043 | nearest_subnormal |
| MNIST | Best loss NVFP4 | 0.021 | 0.009 | E4M3 | 16 | STE | STE | N/A | STE | None_exact | StableSPAM | True | Stochastic | True | absmax | 4.750 | 0.050 | nearest_subnormal |
| MNIST | Pure FP4 | 2.188 | 2.258 | E4M3 | 16 | STE | STE | N/A | STE | None_exact | Adam | False | TiesToEven | False | N/A | 0.000 | -80.086 | to_one |
| MNIST | Pure FP4 | 0.047 | 0.044 | E8M0 | 32 | STE | STE | N/A | STE | None_exact | Adam | False | TiesToEven | False | N/A | 0.000 | -0.738 | nearest_subnormal |
| llama_9M | Baseline | 4.183 | 4.013 | N/A | N/A | N/A | N/A | N/A | N/A | N/A | Adam | False | N/A | N/A | N/A | N/A | N/A | N/A |
| llama_9M | Baseline | 4.141 | 3.972 | N/A | N/A | N/A | N/A | N/A | N/A | N/A | StableSPAM | False | N/A | N/A | N/A | N/A | N/A | N/A |
| llama_9M | Best Score (Neg) | 4.433 | 4.271 | E4M3 | 16 | STE | STE | N/A | STE | IntelFP4_exact | StableSPAM | False | TiesToEven | False | N/A | 0.500 | -0.057 | nearest_subnormal |
| llama_9M | Best Score (Neg) | 4.377 | 4.210 | E8M0 | 32 | STE | STE | N/A | STE | None_exact | StableSPAM | False | TiesToEven | False | N/A | 0.500 | -0.057 | nearest_subnormal |
| llama_9M | Best loss MXFP4 | 4.377 | 4.210 | E8M0 | 32 | STE | STE | N/A | STE | None_exact | StableSPAM | False | TiesToEven | False | N/A | 0.500 | -0.057 | nearest_subnormal |
| llama_9M | Best loss NVFP4 | 4.408 | 4.245 | E4M3 | 16 | STE | STE | N/A | STE | IntelFP4_exact | StableSPAM | True | TiesToEven | True | ignore | 2.000 | -0.129 | nearest_subnormal |
| llama_9M | Pure FP4 | 5.133 | 5.006 | E4M3 | 16 | STE | STE | N/A | STE | None_exact | Adam | False | TiesToEven | False | N/A | 0.000 | -0.239 | to_one |
| llama_9M | Pure FP4 | 4.435 | 4.268 | E8M0 | 32 | STE | STE | N/A | STE | None_exact | Adam | False | TiesToEven | False | N/A | 0.000 | -0.071 | nearest_subnormal |
| small_diffusion | Baseline | 0.029 | 0.029 | N/A | N/A | N/A | N/A | N/A | N/A | N/A | Adam | False | N/A | N/A | N/A | N/A | N/A | N/A |
| small_diffusion | Baseline | 0.019 | 0.019 | N/A | N/A | N/A | N/A | N/A | N/A | N/A | StableSPAM | False | N/A | N/A | N/A | N/A | N/A | N/A |
| small_diffusion | Best Score (Neg) | 0.019 | 0.019 | E4M3 | 16 | STE | STE | N/A | STE | None_exact | StableSPAM | True | TiesToEven | True | ignore | 1.500 | -0.001 | nearest_subnormal |
| small_diffusion | Best Score (Neg) | 0.019 | 0.019 | E8M0 | 32 | STE | STE | all_exact | STE | None_exact | StableSPAM | False | TiesToEven | True | ignore | 2.000 | -0.000 | nearest_subnormal |
| small_diffusion | Best Score (Pos) | 0.018 | 0.019 | E4M3 | 16 | STE | STE | N/A | STE | IntelFP4_exact | StableSPAM | True | TowardPositive | False | N/A | 1.500 | 0.020 | nearest_subnormal |
| small_diffusion | Best Score (Pos) | 0.018 | 0.019 | E8M0 | 32 | STE | STE | N/A | STE | IntelFP4_exact | StableSPAM | False | TiesToEven | False | N/A | 1.000 | 0.028 | nearest_subnormal |
| small_diffusion | Best loss MXFP4 | 0.018 | 0.019 | E4M3 | 16 | STE | STE | N/A | STE | None_exact | StableSPAM | False | TiesToEven | False | N/A | 1.500 | 0.020 | nearest_subnormal |
| small_diffusion | Best loss NVFP4 | 0.018 | 0.019 | E8M0 | 16 | STE | baseline | N/A | STE | IntelFP4_exact | StableSPAM | True | TiesToEven | True | ignore | 4.000 | 0.008 | nearest_subnormal |
| small_diffusion | Pure FP4 | 0.031 | 0.031 | E4M3 | 16 | STE | STE | N/A | STE | None_exact | Adam | False | TiesToEven | False | N/A | 0.000 | -0.656 | nearest_subnormal |
| small_diffusion | Pure FP4 | 0.030 | 0.031 | E8M0 | 32 | STE | STE | N/A | STE | None_exact | Adam | False | TiesToEven | False | N/A | 0.000 | -0.572 | nearest_subnormal |

A.3 TESTING SMOOTHSWIGLU, TENSOR SCALING AND SR

We replicate the results in Chmiel et al. (2025) more exactly by adding the `SmoothSwiGLU` in Fishman et al. (2025). We could not replicate their indicated results on models up to LLama 1B in Figure 6 and Table 6.

Figure 6: Training and validation performance curves LLama with SSwiGLU. The gap between BFLOAT16 still grows with model size depsite tensor scaling and SR.

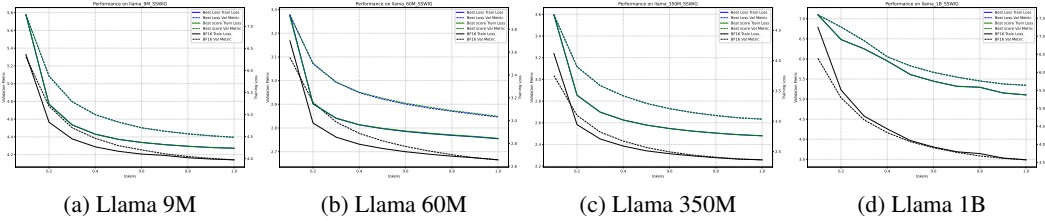

(a) Llama 9M      (b) Llama 60M      (c) Llama 350M      (d) Llama 1B

Table 6: SWIG results

| Dataset | Source | Val loss | Train loss | Scale | Block size | Max grad. | Quant. grad | Hadamard | Scale grad | SR | Optimiser | Loss scaling | Round mode | Tensor scaling | Tensor grad | Complexity points | Score | NaN mode |
|---|---|---|---|---|---|---|---|---|---|---|---|---|---|---|---|---|---|---|
| llama_1B | Baseline | 3.578 | 3.682 | N/A | N/A | N/A | N/A | N/A | N/A | N/A | Adam | False | N/A | N/A | N/A | N/A | N/A | N/A |
| llama_1B | Baseline | 3.487 | 3.569 | N/A | N/A | N/A | N/A | N/A | N/A | N/A | StableSPAM | False | N/A | N/A | N/A | N/A | N/A | N/A |
| llama_1B_SSWIG | Best Score (Neg) | 5.343 | 5.372 | E4M3 | 16 | STE | STE | None_exact | STE | None_exact | StableSPAM | False | TiesToEven | True | ignore | 1.000 | -0.532 | to_one |
| llama_1B_SSWIG | Best loss NVFP4 | 5.343 | 5.372 | E4M3 | 16 | STE | STE | None_exact | STE | None_exact | StableSPAM | False | TiesToEven | True | ignore | 1.000 | -0.532 | to_one |
| llama_350M | Baseline | 2.269 | 2.375 | N/A | N/A | N/A | N/A | N/A | N/A | N/A | Adam | False | N/A | N/A | N/A | N/A | N/A | N/A |
| llama_350M | Baseline | 2.258 | 2.363 | N/A | N/A | N/A | N/A | N/A | N/A | N/A | StableSPAM | False | N/A | N/A | N/A | N/A | N/A | N/A |
| llama_350M_SSWIG | Best Score (Neg) | 2.634 | 2.760 | E4M3 | 16 | STE | STE | None_exact | STE | IntelFP4_exact | StableSPAM | False | TiesToEven | True | ignore | 1.500 | -0.250 | to_one |
| llama_350M_SSWIG | Best loss E4M3 | 2.634 | 2.760 | E4M3 | 16 | STE | STE | None_exact | STE | IntelFP4_exact | StableSPAM | False | TiesToEven | True | ignore | 1.500 | -0.250 | to_one |
| llama_60M | Baseline | 2.665 | 2.657 | N/A | N/A | N/A | N/A | N/A | N/A | N/A | Adam | False | N/A | N/A | N/A | N/A | N/A | N/A |
| llama_60M | Baseline | 2.983 | 3.028 | N/A | N/A | N/A | N/A | N/A | N/A | N/A | StableSPAM | False | N/A | N/A | N/A | N/A | N/A | N/A |
| llama_60M_SSWIG | Best Score (Neg) | 2.849 | 2.845 | E4M3 | 16 | STE | STE | None_exact | STE | IntelFP4_exact | StableSPAM | False | TiesToEven | True | ignore | 1.500 | -0.103 | to_one |
| llama_60M_SSWIG | Best loss NVFP4 | 2.846 | 2.842 | E4M3 | 16 | STE | STE | None_exact | STE | IntelFP4_exact | StableSPAM | True | TiesToEven | True | ignore | 2.000 | -0.136 | to_one |
| llama_9M | Baseline | 4.183 | 4.013 | N/A | N/A | N/A | N/A | N/A | N/A | N/A | Adam | False | N/A | N/A | N/A | N/A | N/A | N/A |
| llama_9M | Baseline | 4.141 | 3.972 | N/A | N/A | N/A | N/A | N/A | N/A | N/A | StableSPAM | False | N/A | N/A | N/A | N/A | N/A | N/A |
| llama_9M_SSWIG | Best Score (Neg) | 4.396 | 4.235 | E4M3 | 16 | STE | STE | None_exact | STE | IntelFP4_exact | StableSPAM | False | TiesToEven | True | ignore | 1.500 | -0.092 | nearest_subnormal |
| llama_9M_SSWIG | Best loss NVFP4 | 4.396 | 4.235 | E4M3 | 16 | STE | STE | None_exact | STE | IntelFP4_exact | StableSPAM | False | TiesToEven | True | ignore | 1.500 | -0.092 | nearest_subnormal |

A.4 CHANGING THE SCALE TO E8M3

During our experiments, we noticed that `E4M3` did not match the performance of `E8M0`, even with tensor scaling. We speculated that the range of `E4M3` was the issue and decided to verify this with an ablation study using `E8M3` to test this hypothesis. We presents the results in Table 7 and Figure 7.

Table 7: E8M3 ablation results

| Dataset | Source | Val loss | Train loss | Scale | Block size | Max grad. | Quant. grad | Hadamard | Scale grad | SR | Optimiser | Loss scaling | Round mode | Tensor scaling | Tensor grad | Complexity points | Score | NaN mode |
|---|---|---|---|---|---|---|---|---|---|---|---|---|---|---|---|---|---|---|
| llama_60M | Baseline | 2.665 | 2.657 | N/A | N/A | N/A | N/A | N/A | N/A | N/A | Adam | False | N/A | N/A | N/A | N/A | N/A | N/A |
| llama_60M | Baseline | 2.983 | 3.028 | N/A | N/A | N/A | N/A | N/A | N/A | N/A | StableSPAM | False | N/A | N/A | N/A | N/A | N/A | nearest_subnormal |
| llama_60M | Best Score (Neg) | 2.775 | 2.773 | E8M3 | 16.000 | STE | STE | None_exact | STE | IntelFP4_exact | StableSPAM | False | TiesToEven | False | N/A | 1.000 | -0.041 | nearest_subnormal |
| llama_60M | Best loss E8M3 | 2.775 | 2.773 | E8M3 | 16.000 | STE | STE | None_exact | STE | IntelFP4_exact | StableSPAM | False | TiesToEven | False | N/A | 1.000 | -0.041 | nearest_subnormal |
| llama_60M | Pure FP4 | 2.851 | 2.848 | E8M3 | 16.000 | STE | STE | None_exact | STE | None_exact | Adam | False | TiesToEven | False | N/A | 0.000 | -0.070 | nearest_subnormal |
| llama_9M | Baseline | 4.183 | 4.013 | N/A | N/A | N/A | N/A | N/A | N/A | N/A | Adam | False | N/A | N/A | N/A | N/A | N/A | N/A |
| llama_9M | Baseline | 4.141 | 3.972 | N/A | N/A | N/A | N/A | N/A | N/A | N/A | StableSPAM | False | N/A | N/A | N/A | N/A | N/A | N/A |
| llama_9M | Best Score (Neg) | 4.271 | 4.106 | E8M3 | 16.000 | STE | STE | None_exact | STE | IntelFP4_exact | StableSPAM | False | TiesToEven | False | N/A | 1.000 | -0.031 | nearest_subnormal |
| llama_9M | Best loss E8M3 | 4.271 | 4.106 | E8M3 | 16.000 | STE | STE | None_exact | STE | IntelFP4_exact | StableSPAM | False | TiesToEven | False | N/A | 1.000 | -0.031 | nearest_subnormal |
| llama_9M | Pure FP4 | 4.320 | 4.156 | E8M3 | 16.000 | STE | STE | None_exact | STE | None_exact | Adam | False | TiesToEven | False | N/A | 0.000 | -0.043 | nearest_subnormal |

Confirming that the limiting factor of `E4M3` is the range, we next speculate that an 8-bit numerical scaling format in-between `E4M3` and `E8M0` might offer a good trade-off between range and precision.

A.5 UE5M3 RESULTS

We present the `UE5M3` experiments in Table 8. We overall find that the `UE5M3` outperforms `MXFP4` when tensor scaling is applied. It should be noted that `UE5M3` will not work without any adjustments unlike `MXFP4`, implying that increased precision often comes with increased overhead. We visualise the training and validation curves in Figure 8 and Figure 9. We further provide the Pareto-frontier plots in Figure 10, we note that generally lower complexity configurations achieve better scores.

Figure 7: Training and validation performance curves for E8M3.

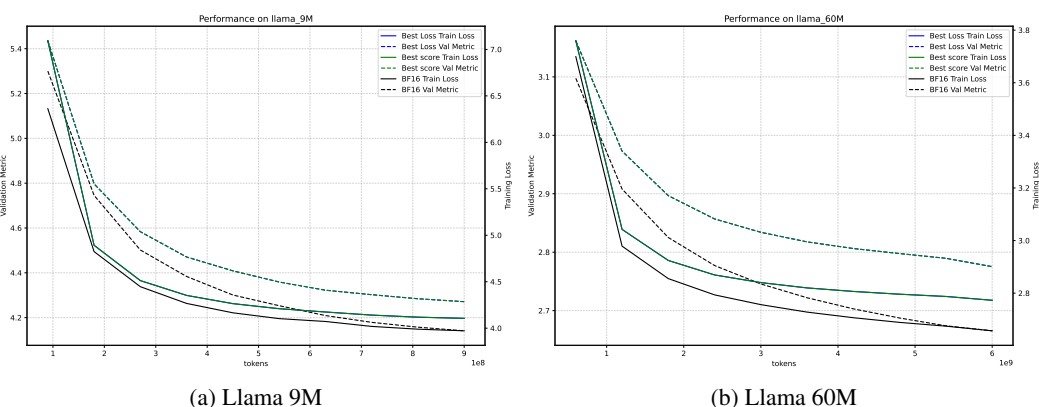

(a) Llama 9M

(b) Llama 60M

Table 8: UE5M3 results

| Dataset | Source | Val loss | Train loss | Scale | Block size | Max grad. | Quant. grad | Hadamard | Scale grad | SR | Optimiser | Loss scaling | Round mode | Tensor scaling | Tensor grad | Complexity points | Score | NaN mode |
|---|---|---|---|---|---|---|---|---|---|---|---|---|---|---|---|---|---|---|
| CIFAR10 | Baseline | 0.875 | 0.003 | N/A | N/A | N/A | N/A | N/A | N/A | N/A | Adam | False | N/A | N/A | N/A | N/A | N/A | N/A |
| CIFAR10 | Baseline | 0.895 | 0.027 | N/A | N/A | N/A | N/A | N/A | N/A | N/A | StableSPAM | False | N/A | N/A | N/A | N/A | N/A | N/A |
| CIFAR10 | Best Score (Neg) | 0.883 | 0.003 | E5M3 | 32 | STE | STE | None_exact | STE | IntelFP4_exact | Adam | True | TiesToEven | False | N/A | 1.000 | -0.009 | nearest_subnormal |
| CIFAR10 | Best Score (Pos) | 0.867 | 0.003 | E5M3 | 32 | STE | STE | None_exact | STE | IntelFP4_exact | Adam | False | TiesToEven | False | N/A | 0.500 | 0.009 | nearest_subnormal |
| CIFAR10 | Best loss E5M3 | 0.867 | 0.003 | E5M3 | 32 | STE | STE | None_exact | STE | IntelFP4_exact | Adam | False | TiesToEven | False | N/A | 0.500 | 0.009 | nearest_subnormal |
| CIFAR10 | Pure FP4 | 0.875 | 0.005 | E5M2 | 32 | STE | STE | None_exact | STE | None_exact | Adam | False | TowardPositive | False | N/A | 0.000 | 0.000 | nearest_subnormal |
| CIFAR10 | Pure FP4 | 1.328 | 1.150 | E5M3 | 32 | STE | STE | None_exact | STE | None_exact | Adam | False | TiesToEven | False | N/A | 0.000 | -0.518 | to_one |
| IMAGENET100 | Baseline | 1.383 | 0.014 | N/A | N/A | N/A | N/A | N/A | N/A | N/A | Adam | False | N/A | N/A | N/A | N/A | N/A | N/A |
| IMAGENET100 | Baseline | 1.750 | 0.078 | N/A | N/A | N/A | N/A | N/A | N/A | N/A | StableSPAM | False | N/A | N/A | N/A | N/A | N/A | N/A |
| IMAGENET100 | Best Score (Neg) | 1.391 | 0.015 | E5M3 | 32 | STE | STE | None_exact | STE | all_activation_exact | Adam | True | TowardPositive | False | N/A | 1.000 | -0.006 | nearest_subnormal |
| IMAGENET100 | Best Score (Pos) | 1.344 | 0.014 | E5M3 | 32 | STE | STE | None_exact | STE | IntelFP4_exact | Adam | False | TiesToEven | False | N/A | 0.500 | 0.028 | nearest_subnormal |
| IMAGENET100 | Best loss E5M3 | 1.344 | 0.014 | E5M3 | 32 | STE | STE | None_exact | STE | IntelFP4_exact | Adam | False | TiesToEven | False | N/A | 0.500 | 0.028 | nearest_subnormal |
| IMAGENET100 | Pure FP4 | 2.031 | 1.530 | E5M3 | 32 | STE | STE | None_exact | STE | None_exact | Adam | False | TowardPositive | False | N/A | 0.000 | -0.469 | nearest_subnormal |
| MNIST | Baseline | 0.027 | 0.016 | N/A | N/A | N/A | N/A | N/A | N/A | N/A | Adam | False | N/A | N/A | N/A | N/A | N/A | N/A |
| MNIST | Baseline | 0.028 | 0.004 | N/A | N/A | N/A | N/A | N/A | N/A | N/A | StableSPAM | False | N/A | N/A | N/A | N/A | N/A | N/A |
| MNIST | Best Score (Neg) | 0.027 | 0.010 | E5M3 | 32 | STE | STE | None_exact | STE | None_exact | StableSPAM | False | TiesToEven | True | ignore | 1.000 | -0.005 | nearest_subnormal |
| MNIST | Best Score (Pos) | 0.025 | 0.006 | E5M3 | 32 | STE | STE | None_exact | STE | IntelFP4_exact | StableSPAM | False | Stochastic | False | N/A | 1.250 | 0.047 | nearest_subnormal |
| MNIST | Best loss E5M3 | 0.025 | 0.006 | E5M3 | 32 | STE | STE | None_exact | STE | IntelFP4_exact | StableSPAM | False | Stochastic | False | N/A | 1.250 | 0.047 | nearest_subnormal |
| MNIST | Pure FP4 | 0.029 | 0.022 | E5M2 | 16 | STE | STE | None_exact | STE | None_exact | Adam | False | TowardPositive | False | N/A | 0.000 | -0.059 | nearest_subnormal |
| MNIST | Pure FP4 | 0.029 | 0.023 | E5M3 | 32 | STE | STE | None_exact | STE | None_exact | Adam | False | TowardPositive | False | N/A | 0.000 | -0.072 | nearest_subnormal |
| big_diffusion | Baseline | 0.135 | 0.128 | N/A | N/A | N/A | N/A | N/A | N/A | N/A | Adam | False | N/A | N/A | N/A | N/A | N/A | N/A |
| big_diffusion | Baseline | 0.113 | 0.110 | N/A | N/A | N/A | N/A | N/A | N/A | N/A | StableSPAM | False | N/A | N/A | N/A | N/A | N/A | N/A |
| big_diffusion | Best Score (Neg) | 0.113 | 0.109 | E5M3 | 32 | STE | STE | None_exact | STE | None_exact | Adam | True | Stochastic | False | N/A | 0.750 | -0.006 | to_one |
| big_diffusion | Best Score (Pos) | 0.104 | 0.100 | E5M3 | 32 | STE | STE | None_exact | STE | all_activation_exact | Adam | False | TowardPositive | False | N/A | 1.000 | 0.074 | to_one |
| big_diffusion | Best loss E5M3 | 0.102 | 0.097 | E5M3 | 32 | STE | STE | None_exact | STE | all_activation_exact | Adam | False | TowardPositive | False | N/A | 1.500 | 0.062 | to_one |
| big_diffusion | Pure FP4 | 0.130 | 0.123 | E5M3 | 32 | STE | STE | None_exact | STE | None_exact | Adam | False | TiesToEven | False | N/A | 0.000 | -0.155 | nearest_subnormal |
| gaussian_reg | Baseline | 25.250 | 25.224 | N/A | N/A | N/A | N/A | N/A | N/A | N/A | Adam | False | N/A | N/A | N/A | N/A | N/A | N/A |
| gaussian_reg | Baseline | 0.013 | 0.013 | N/A | N/A | N/A | N/A | N/A | N/A | N/A | StableSPAM | False | N/A | N/A | N/A | N/A | N/A | N/A |
| gaussian_reg | Best Score (Neg) | 25.875 | 26.250 | E5M3 | 32 | STE | STE | None_exact | STE | None_exact | StableSPAM | False | TiesToEven | True | ignore | 1.000 | -1998.698 | nearest_subnormal |
| gaussian_reg | Best loss E5M3 | 25.250 | 26.237 | E5M3 | 32 | STE | STE | IntelFP4_exact | STE | IntelFP4_exact | StableSPAM | True | Stochastic | True | ignore | 3.250 | -6338.788 | nearest_subnormal |
| gaussian_reg | Pure FP4 | 30.000 | 29.974 | E5M2 | 16 | STE | STE | backward_exact | STE | None_exact | Adam | False | TiesToEven | False | N/A | 0.000 | -2317.491 | nearest_subnormal |
| gaussian_reg | Pure FP4 | 31.375 | 31.836 | E5M3 | 32 | STE | STE | None_exact | STE | None_exact | Adam | False | TiesToEven | False | N/A | 0.000 | -2423.755 | nearest_subnormal |
| llama_1B | Baseline | 3.578 | 3.682 | N/A | N/A | N/A | N/A | N/A | N/A | N/A | Adam | False | N/A | N/A | N/A | N/A | N/A | N/A |
| llama_1B | Baseline | 3.487 | 3.569 | N/A | N/A | N/A | N/A | N/A | N/A | N/A | StableSPAM | False | N/A | N/A | N/A | N/A | N/A | N/A |
| llama_1B | Best Score (Neg) | 3.586 | 3.666 | E5M3 | 32 | STE | STE | None_exact | STE | IntelFP4_exact | StableSPAM | False | TiesToEven | True | ignore | 1.500 | -0.043 | to_one |
| llama_1B | Best loss E5M3 | 3.586 | 3.666 | E5M3 | 32 | STE | STE | None_exact | STE | IntelFP4_exact | StableSPAM | False | TiesToEven | True | ignore | 1.500 | -0.043 | to_one |
| llama_1B | Pure FP4 | 6.830 | 6.802 | E5M3 | 32 | STE | STE | None_exact | STE | None_exact | StableSPAM | False | TiesToEven | False | N/A | 0.000 | -0.959 | to_one |
| llama_350M | Baseline | 2.269 | 2.375 | N/A | N/A | N/A | N/A | N/A | N/A | N/A | Adam | False | N/A | N/A | N/A | N/A | N/A | N/A |
| llama_350M | Baseline | 2.258 | 2.363 | N/A | N/A | N/A | N/A | N/A | N/A | N/A | StableSPAM | False | N/A | N/A | N/A | N/A | N/A | N/A |
| llama_350M | Best Score (Neg) | 2.322 | 2.437 | E5M3 | 32 | STE | STE | None_exact | STE | IntelFP4_exact | StableSPAM | False | TiesToEven | True | ignore | 1.500 | -0.043 | to_one |
| llama_350M | Best loss E5M3 | 2.322 | 2.437 | E5M3 | 32 | STE | STE | None_exact | STE | IntelFP4_exact | StableSPAM | False | TiesToEven | True | ignore | 1.500 | -0.043 | to_one |
| llama_350M | Pure FP4 | 4.884 | 4.963 | E5M3 | 32 | STE | STE | None_exact | STE | None_exact | StableSPAM | False | TiesToEven | False | N/A | 0.000 | -1.163 | to_one |
| llama_60M | Baseline | 2.665 | 2.657 | N/A | N/A | N/A | N/A | N/A | N/A | N/A | Adam | False | N/A | N/A | N/A | N/A | N/A | N/A |
| llama_60M | Baseline | 2.983 | 3.028 | N/A | N/A | N/A | N/A | N/A | N/A | N/A | StableSPAM | False | N/A | N/A | N/A | N/A | N/A | N/A |
| llama_60M | Best Score (Neg) | 2.791 | 2.788 | E5M3 | 32 | STE | STE | None_exact | STE | IntelFP4_exact | StableSPAM | False | TiesToEven | True | ignore | 1.500 | -0.071 | to_one |
| llama_60M | Best loss E5M3 | 2.791 | 2.788 | E5M3 | 32 | STE | STE | None_exact | STE | IntelFP4_exact | StableSPAM | False | TiesToEven | True | ignore | 1.500 | -0.071 | to_one |
| llama_60M | Pure FP4 | 5.056 | 5.050 | E5M3 | 32 | STE | STE | None_exact | STE | None_exact | StableSPAM | False | TiesToEven | False | N/A | 0.000 | -0.897 | to_one |
| llama_9M | Baseline | 4.183 | 4.013 | N/A | N/A | N/A | N/A | N/A | N/A | N/A | Adam | False | N/A | N/A | N/A | N/A | N/A | N/A |
| llama_9M | Baseline | 4.141 | 3.972 | N/A | N/A | N/A | N/A | N/A | N/A | N/A | StableSPAM | False | N/A | N/A | N/A | N/A | N/A | N/A |
| llama_9M | Best Score (Neg) | 4.290 | 4.125 | E5M3 | 32 | STE | STE | None_exact | STE | IntelFP4_exact | StableSPAM | False | TiesToEven | True | ignore | 1.000 | -0.036 | to_one |
| llama_9M | Best loss E5M3 | 4.280 | 4.115 | E5M3 | 32 | STE | STE | None_exact | STE | IntelFP4_exact | StableSPAM | True | TiesToEven | True | ignore | 2.000 | -0.067 | to_one |
| llama_9M | Pure FP4 | 5.437 | 5.332 | E5M3 | 32 | STE | STE | None_exact | STE | None_exact | StableSPAM | False | TiesToEven | False | N/A | 0.000 | -0.313 | to_one |
| small_diffusion | Baseline | 0.029 | 0.029 | N/A | N/A | N/A | N/A | N/A | N/A | N/A | Adam | False | N/A | N/A | N/A | N/A | N/A | N/A |
| small_diffusion | Baseline | 0.019 | 0.019 | N/A | N/A | N/A | N/A | N/A | N/A | N/A | StableSPAM | False | N/A | N/A | N/A | N/A | N/A | N/A |
| small_diffusion | Best Score (Neg) | 0.019 | 0.019 | E5M3 | 32 | STE | STE | None_exact | STE | IntelFP4_exact | StableSPAM | False | TiesToEven | True | ignore | 1.000 | -0.000 | to_one |
| small_diffusion | Best Score (Pos) | 0.018 | 0.019 | E5M3 | 32 | STE | STE | None_exact | STE | IntelFP4_exact | StableSPAM | False | TiesToEven | False | N/A | 1.000 | 0.027 | to_one |
| small_diffusion | Best loss E5M3 | 0.018 | 0.019 | E5M3 | 32 | STE | STE | None_exact | STE | IntelFP4_exact | StableSPAM | True | TiesToEven | False | N/A | 1.500 | 0.021 | nearest_subnormal |
| small_diffusion | Pure FP4 | 0.023 | 0.024 | E5M3 | 32 | STE | STE | None_exact | STE | None_exact | Adam | False | TiesToEven | False | N/A | 0.000 | -0.233 | to_one |

Figure 8: Training and validation performance curves for selected models and datasets of `UE5M3` experiments.

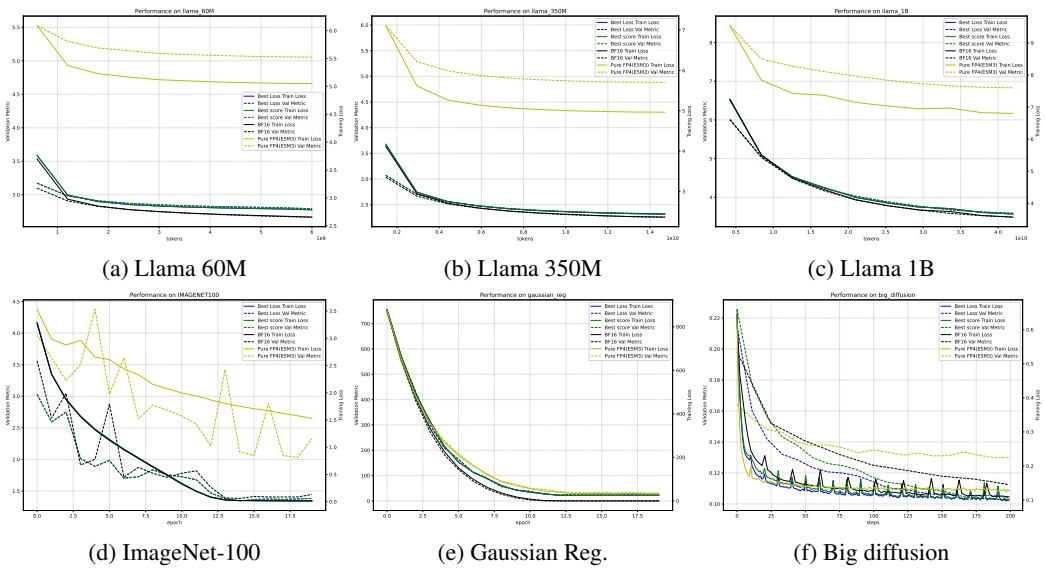

(a) Llama 60M        (b) Llama 350M        (c) Llama 1B

(d) ImageNet-100        (e) Gaussian Reg.        (f) Big diffusion

Figure 9: Training and validation performance curves for additional dataset for the `UE5M3` scale

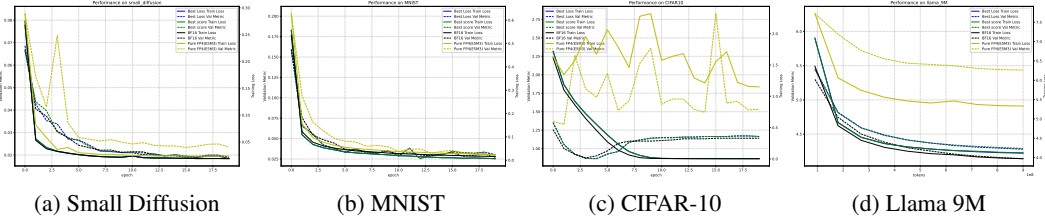

(a) Small Diffusion      (b) MNIST      (c) CIFAR-10      (d) Llama 9M

Figure 10: Pareto-frontier plots for each dataset, UE5M3 results. Note that we swept a smaller space compared to the main experiments.

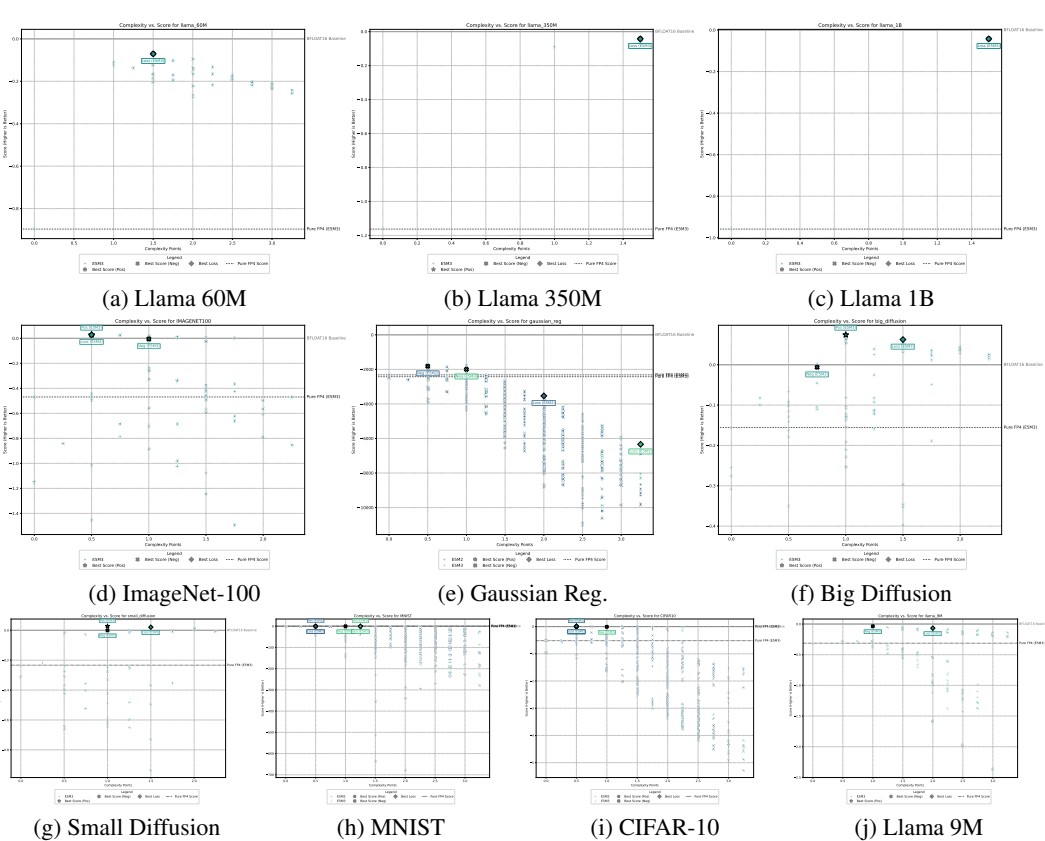

(a) Llama 60M  (b) Llama 350M  (c) Llama 1B

(d) ImageNet-100  (e) Gaussian Reg.  (f) Big Diffusion

(g) Small Diffusion  (h) MNIST  (i) CIFAR-10  (j) Llama 9M

### A.6 EXPERIMENTATION DETAILS

**Complexity Score Calculation** The complexity penalty $\Omega(c)$ is calculated based on the set of techniques $\mathcal{T}$ used in a configuration. The total set of techniques and their corresponding weights $w_t$ are detailed in Table 9. A configuration's total complexity is the sum of weights for all techniques it employs, i.e., $\Omega(c) = \sum_{t \in \mathcal{T}_c} w_t$, where $\mathcal{T}_c \subseteq \mathcal{T}$.

Table 9: Complexity weights for non-baseline techniques.

| Technique ($t$) | Activation Condition | Weight ($w_t$) |
|---|---|---|
| Non-STE Smoothing | `smooth` $\neq$ 'STE' | 3.0 |
| Tensor Scaling Gradient Est. | `tensor_scaling_grad_est` is active | 3.0 |
| Non-STE Step Gradient | `stepGradient` $\neq$ 'STE' | 2.0 |
| Hadamard Transform | `use_hadamard` is active | 1.0 |
| Non-STE Quantized Gradient | `qGradient` $\neq$ 'STE' | 1.5 |
| Stochastic Rounding (SR) | `SR` is active | 0.5 |
| Tensor Scaling | `use_tensor_scaling` is active | 0.5 |
| Loss Scaling | `loss_scaling` is True | 0.5 |
| SPAM Optimizer | 'SPAM' in `optimiser` name | 0.5 |
| Stochastic Rounding for scale | Scale rounding is `Stochastic` | 0.25 |

We motivate the weight with reference to the added complexity and memory overhead, and fusability on a lower level language based on Table 2.

Table 10: Summary of the Full Hyperparameter Sweep.

| Group | Parameter | Search Values |
|---|---|---|
| Quant. | Scale Format | {E8M0, E4M3} |
| | Max Approx. | {STE, softsoftmax, hardsoftmax, absmax} |
| | Scale Rounding | {TiesToEven, TowardPositive, SR} |
| Gradient | Step Gradient | {STE, baseline, spline} |
| | Scaling Quant. [1] | {STE, baseline, spline} |
| | Tensor Scale Grad.[2] | {ignore, absmax, STE} |
| Opt. | Optimizer | {Adam, StableSPAM} |
| | Loss Scaling | {True, False} |
| | Tensor Scaling | {True, False} |
| | SR | {None, all act., backward act.} |
| | Hadamard | {None, all, backward} |

[1]**Conditional**: Options for `Scaling Quant.` depend on the values of `Max Approx.`
[2]**Conditional**: Options for `Tensor Scale Grad.` depend on the values of `Tensor Scaling` and `Max Approx.`

**Parameter sweeps**

**Dataset descriptions**

Table 11: Summary of Experimental Setups

| Learning Task | Dataset / Model | Global Batch Size | Seq. Length | Grad. Accum. | Learning Rate | Training Duration |
|---|---|---|---|---|---|---|
| Regression | Synthetic Gaussian | 4096 | | - | $1 \times 10^{-2}$ | 20 Epochs |
| Classification | MNIST | 512 | | - | $1 \times 10^{-3}$ | 20 Epochs |
| Classification | CIFAR-10 | 512 | | - | $1 \times 10^{-3}$ | 20 Epochs |
| Classification | ImageNet-100 | 512 | | - | $1 \times 10^{-3}$ | 20 Epochs |
| Image Generation | CIFAR-10 (Small U-Net, small_diffusion) | 512 | | - | $1 \times 10^{-3}$ | 20 Epochs |
| Image Generation | FFHQ (Big U-Net, big_diffusion) | 20 | | - | $1 \times 10^{-4}$ | 3 Epochs |
| Language Modeling | LLaMA-9M | 4096 | 128 | 1 | $1 \times 10^{-3}$ | 0.9B Tokens[*] |
| Language Modeling | LLaMA-60M | 128 | 512 | 1 | $1 \times 10^{-4}$ | 6B Tokens[*] |
| Language Modeling | LLaMA-350M | 16 | 1024 | 8 | $1 \times 10^{-4}$ | 14.7B Tokens[*] |
| Language Modeling | LLaMA-1B | 4 | 1024 | 512 | $1 \times 10^{-4}$ | 42B Tokens[*] |

[*]We use the WikiText dataset. Training duration is calculated based on parameter count (100x for models <350M, $\approx 42\times$ otherwise). We chose the token count based on Fishman et al. (2025), which show that training divergence in low precision usually happen around this amount of tokens relative to model size. Gradient accumulation is used for the 350M and 1B models. The 350M and 1B model configurations where taken directly from Tseng et al. (2025).

## A.7 DERIVATION OF PROPOSITION 1

The function for a single element is:

$$f_{ij} = \frac{1}{s_q} Q(s_q \mathbf{X}_{ij})$$

Since $s_q$ is a function of $\mathbf{X}_{ij}$, we must use the product rule on: $\left(\frac{1}{s_q}\right)$ and $(Q(s_q \cdot \mathbf{X}_{ij}))$.

$$\frac{\partial f_{ij}}{\partial \mathbf{X}_{ij}} = \left(\frac{\partial}{\partial \mathbf{X}_{ij}} \frac{1}{s_q}\right) \cdot Q(s_q \mathbf{X}_{ij}) + \frac{1}{s_q} \cdot \left(\frac{\partial}{\partial \mathbf{X}_{ij}} Q(s_q \mathbf{X}_{ij})\right)$$

Term 1: Derivative of $\frac{1}{s_q}$ This derivative depends on how $s_q$ is defined by $\mathbf{X}$. Let $s = s(\mathbf{X})$.

$$\frac{\partial}{\partial \mathbf{X}_{ij}} \left(\frac{1}{s_q}\right) = -\frac{1}{s_q^2} \frac{\partial s_q}{\partial \mathbf{X}_{ij}} = -\frac{q'(s)}{s_q^2} \frac{\partial s}{\partial \mathbf{X}_{ij}}$$

Term 2: Derivative of $Q(s_q \mathbf{X}_{ij})$ We apply the chain rule to $Q$, and then the product rule to its argument $(s_q \mathbf{X}_{ij})$.

$$\frac{\partial}{\partial \mathbf{X}_{ij}} Q(s_q \mathbf{X}_{ij}) = Q'(s_q \mathbf{X}_{ij}) \cdot \frac{\partial(s_q \mathbf{X}_{ij})}{\partial \mathbf{X}_{ij}}$$

$$\frac{\partial(s_q \mathbf{X}_{ij})}{\partial \mathbf{X}_{ij}} = \left(\frac{\partial s_q}{\partial \mathbf{X}_{ij}}\right) \mathbf{X}_{ij} + s_q \left(\frac{\partial \mathbf{X}_{ij}}{\partial \mathbf{X}_{ij}}\right) = \mathbf{X}_{ij} \frac{\partial s_q}{\partial \mathbf{X}_{ij}} + s_q = \mathbf{X}_{ij} q'(s) \frac{\partial s}{\partial \mathbf{X}_{ij}} + s_q$$

Thus, the full derivative of the second term is:

$$\frac{\partial}{\partial \mathbf{X}_{ij}} Q(s_q \mathbf{X}_{ij}) = Q'(s_q \mathbf{X}_{ij}) \left(\mathbf{X}_{ij} q'(s) \frac{\partial s}{\partial \mathbf{X}_{ij}} + s_q\right)$$

**Combining and Final Result**   We substitute the results for both terms back into the main equation:

$$\frac{\partial f_{ij}}{\partial \mathbf{X}_{ij}} = \left(-\frac{q'(s)}{s_q^2} \frac{\partial s}{\partial \mathbf{X}_{ij}}\right) Q(s_q \mathbf{X}_{ij}) + \frac{1}{s_q} \left[Q'(s_q \mathbf{X}_{ij}) \left(\mathbf{X}_{ij} q'(s) \frac{\partial s}{\partial \mathbf{X}_{ij}} + s_q\right)\right]$$

Distributing the $\frac{1}{s_q}$ term:

$$\frac{\partial f_{ij}}{\partial \mathbf{X}_{ij}} = -\frac{q'(s)}{s_q^2} Q(s_q \mathbf{X}_{ij}) \frac{\partial s}{\partial \mathbf{X}_{ij}} + \frac{\mathbf{X}_{ij}}{s_q} Q'(s_q \mathbf{X}_{ij}) q'(s) \frac{\partial s}{\partial \mathbf{X}_{ij}} + \frac{s_q}{s_q} Q'(s_q \mathbf{X}_{ij})$$

Grouping the terms by their derivative component gives the final result for this fully general model:

$$\boxed{\frac{\partial f_{ij}}{\partial \mathbf{X}_{ij}} = Q'(s_q \mathbf{X}_{ij}) + \frac{\partial s}{\partial \mathbf{X}_{ij}} \left[\frac{q'(s)}{s_q} \left(\mathbf{X}_{ij} Q'(s_q \mathbf{X}_{ij}) - \frac{1}{s_q} Q(s_q \mathbf{X}_{ij})\right)\right]} \tag{6}$$

## A.8 THEOREM 2 DERIVATION

*Proof.* We want to find the partial derivative of $h_{ij}(\mathbf{X})$ with respect to an element $\mathbf{X}_{ij}$. The transformation is defined as:

$$h_{ij}(\mathbf{X}) = g(\mathbf{X}) \cdot f_{ij}(\mathbf{U}_p)$$

where $g(\mathbf{X}) = \text{absmax}(\mathbf{X})$ and $\mathbf{U}_p = \mathbf{X}_p / g(\mathbf{X})$. An element $\mathbf{X}_{ij}$ belongs to a specific block $p$.

1. **Apply the Product Rule.** We treat $g$ and $f_{ij}$ as two functions of $\mathbf{X}$. The product rule states $(uv)' = u'v + uv'$.

$$\frac{\partial h_{ij}}{\partial \mathbf{X}_{ij}} = \frac{\partial g}{\partial \mathbf{X}_{ij}} \cdot f_{ij}(\mathbf{U}_p) + g \cdot \frac{\partial f_{ij}(\mathbf{U}_p)}{\partial \mathbf{X}_{ij}}$$

2. **Apply the Chain Rule.** The second term's derivative requires the chain rule because $f_{ij}$ is a function of $\mathbf{U}_{p,ij}$, which is a function of $\mathbf{X}_{ij}$.

$$\frac{\partial f_{ij}(\mathbf{U}_p)}{\partial \mathbf{X}_{ij}} = \frac{\partial f_{ij}}{\partial \mathbf{U}_{p,ij}} \cdot \frac{\partial \mathbf{U}_{p,ij}}{\partial \mathbf{X}_{ij}}$$

3. **Apply the Quotient Rule.** We find the derivative of $\mathbf{U}_{p,ij} = \mathbf{X}_{ij}/g$ with respect to $\mathbf{X}_{ij}$ using the quotient rule $(\frac{u}{v})' = \frac{u'v - uv'}{v^2}$.

$$\frac{\partial \mathbf{U}_{p,ij}}{\partial \mathbf{X}_{ij}} = \frac{1 \cdot g - \mathbf{X}_{ij} \cdot \frac{\partial g}{\partial \mathbf{X}_{ij}}}{g^2} = \frac{1}{g} - \frac{\mathbf{X}_{ij}}{g^2}\frac{\partial g}{\partial \mathbf{X}_{ij}}$$

4. **Substitute and Combine.** Now, substitute the result from step (3) into step (2), and then the result of that into step (1).

$$\frac{\partial h_{ij}}{\partial \mathbf{X}_{ij}} = \frac{\partial g}{\partial \mathbf{X}_{ij}}f_{ij}(\mathbf{U}_p) + g \cdot \left[\frac{\partial f_{ij}}{\partial \mathbf{U}_{p,ij}}\left(\frac{1}{g} - \frac{\mathbf{X}_{ij}}{g^2}\frac{\partial g}{\partial \mathbf{X}_{ij}}\right)\right]$$

5. **Simplify and Rearrange.** Distribute the outer $g$ into the brackets.

$$\frac{\partial h_{ij}}{\partial \mathbf{X}_{ij}} = \frac{\partial g}{\partial \mathbf{X}_{ij}}f_{ij}(\mathbf{U}_p) + \frac{g}{g}\frac{\partial f_{ij}}{\partial \mathbf{U}_{p,ij}} - \frac{g \cdot \mathbf{X}_{ij}}{g^2}\frac{\partial f_{ij}}{\partial \mathbf{U}_{p,ij}}\frac{\partial g}{\partial \mathbf{X}_{ij}}$$

The terms simplify, and we can replace $\frac{\mathbf{X}_{ij}}{g}$ with its definition, $\mathbf{U}_{p,ij}$.

$$\frac{\partial h_{ij}}{\partial \mathbf{X}_{ij}} = \frac{\partial g}{\partial \mathbf{X}_{ij}}f_{ij}(\mathbf{U}_p) + \frac{\partial f_{ij}}{\partial \mathbf{U}_{p,ij}} - \mathbf{U}_{p,ij}\frac{\partial f_{ij}}{\partial \mathbf{U}_{p,ij}}\frac{\partial g}{\partial \mathbf{X}_{ij}}$$

Finally, we group the terms containing $\frac{\partial g}{\partial \mathbf{X}_{ij}}$ to arrive at the theorem's statement.

$$\frac{\partial h_{ij}}{\partial \mathbf{X}_{ij}} = \frac{\partial f_{ij}}{\partial \mathbf{U}_{p,ij}} + \frac{\partial g}{\partial \mathbf{X}_{ij}}\left(f_{ij}(\mathbf{U}_p) - \mathbf{U}_{p,ij}\frac{\partial f_{ij}}{\partial \mathbf{U}_{p,ij}}\right)$$

This completes the proof. $\qquad\square$

### A.9 PROPOSTION 1 PROOF

*Proof.* The result follows directly from applying the chain rule to $s(Z(\mathbf{X}))$.

$$\frac{\partial s}{\partial \mathbf{X}_{ij}} = \frac{ds}{dZ} \cdot \frac{\partial Z}{\partial \mathbf{X}_{ij}} = \frac{d}{dZ}\left(\frac{\texttt{FP4 max}}{Z}\right)\frac{\partial Z}{\partial \mathbf{X}_{ij}} = -\frac{\texttt{FP4 max}}{Z(\mathbf{X})^2}\frac{\partial Z}{\partial \mathbf{X}_{ij}}$$

$\qquad\square$

### A.10 ABSMAX GRADIENT DERIVATION

*Proof.* Let $(i^*, j^*)$ be the index of the element with the maximum absolute value, such that $Z(\mathbf{X}) = |\mathbf{X}_{i^*j^*}|$. We first find the gradient of $Z(\mathbf{X})$. The derivative of the absolute value function is the sign function, $\frac{d|x|}{dx} = \text{sign}(x)$. The derivative is non-zero only when we differentiate with respect to the element $\mathbf{X}_{i^*j^*}$ itself. This can be expressed precisely using the Kronecker delta:

$$\frac{\partial Z}{\partial \mathbf{X}_{ij}} = \frac{\partial |\mathbf{X}_{i^*j^*}|}{\partial \mathbf{X}_{ij}} = \text{sign}(\mathbf{X}_{i^*j^*}) \cdot \delta_{ii^*}\delta_{jj^*}$$

Substituting this result into the formula from Theorem 2 completes the proof.

$$\frac{\partial s}{\partial \mathbf{X}_{ij}} = -\frac{\text{FP4 max}}{Z(\mathbf{X})^2}\frac{\partial Z}{\partial \mathbf{X}_{ij}} = -\frac{\text{FP4 max}}{Z(\mathbf{X})^2}\left(\text{sign}(\mathbf{X}_{i^*j^*}) \cdot \delta_{ii^*}\delta_{jj^*}\right)$$

$\qquad\square$

### A.11 SOFTMAX GRADIENT DERIVATION

*Proof.* We first find the gradient of $Z(\mathbf{X})$ by applying the chain rule multiple times.

$$\frac{\partial Z}{\partial \mathbf{X}_{ij}} = \frac{\partial}{\partial \mathbf{X}_{ij}} \left[ \frac{1}{\beta} \log \left( \sum_{k,l} e^{\beta |\mathbf{X}_{kl}|} \right) \right]$$

$$= \frac{1}{\beta} \cdot \frac{1}{\sum_{k,l} e^{\beta |\mathbf{X}_{kl}|}} \cdot \frac{\partial}{\partial \mathbf{X}_{ij}} \left( e^{\beta |\mathbf{X}_{ij}|} \right)$$

$$= \frac{1}{\beta} \cdot \frac{1}{\sum_{k,l} e^{\beta |\mathbf{X}_{kl}|}} \cdot \left( e^{\beta |\mathbf{X}_{ij}|} \cdot \beta \cdot \operatorname{sign}(\mathbf{X}_{ij}) \right)$$

$$= \frac{e^{\beta |\mathbf{X}_{ij}|}}{\sum_{k,l} e^{\beta |\mathbf{X}_{kl}|}} \cdot \operatorname{sign}(\mathbf{X}_{ij})$$

The fractional term is the definition of the softmax function applied to the scaled, absolute values of the tensor elements. Thus:

$$\frac{\partial Z}{\partial \mathbf{X}_{ij}} = \operatorname{softmax}(\beta |\mathbf{X}|)_{ij} \cdot \operatorname{sign}(\mathbf{X}_{ij})$$

Substituting this dense gradient back into the formula from Theorem 2 completes the proof.

$$\frac{\partial s}{\partial \mathbf{X}_{ij}} = -\frac{\mathtt{FP4\ max}}{Z(\mathbf{X})^2} \frac{\partial Z}{\partial \mathbf{X}_{ij}} = -\frac{\mathtt{FP4\ max}}{Z(\mathbf{X})^2} \left( \operatorname{softmax}(\beta |\mathbf{X}|)_{ij} \cdot \operatorname{sign}(\mathbf{X}_{ij}) \right)$$

$\square$

### A.12 TENSOR RECONSTRUCTION ERROR WITH MXFP4 FORMAT

**Reconstruction error** We first consider the reconstruction error, i.e., $|\mathbf{X} - \frac{1}{s_q} Q(s_q \cdot \mathbf{X})|$ for different choices of $k$, rounding modes of $s$, block sizes, and max functions $Z(\mathbf{X})$. We illustrate different slices of the relative error $\frac{|\mathbf{X} - \frac{1}{s_q} Q(s_q \cdot \mathbf{X})|}{|\mathbf{X}|}$. Figure 11 shows the reconstruction error for the Straight-Through Estimator (STE) as a function of block size. As expected, the error decreases as the block size increases.

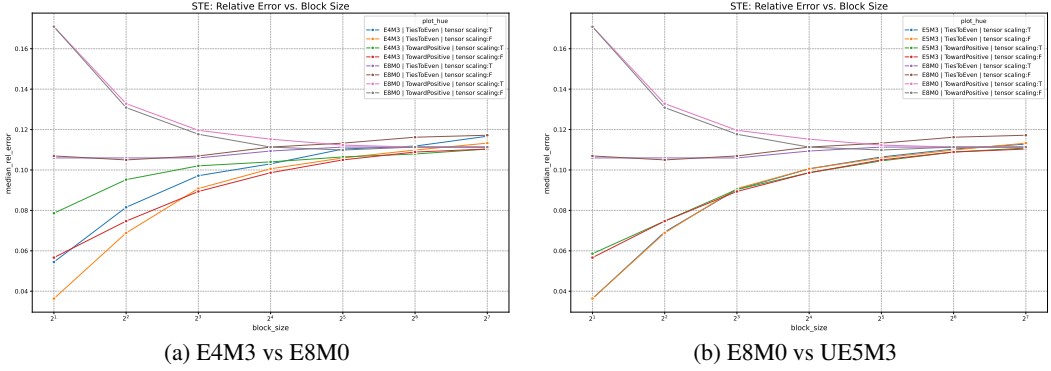

(a) E4M3 vs E8M0          (b) E8M0 vs UE5M3

Figure 11: Reconstruction error using STE as a function of block size.

**Experiment 2: STE Error vs. Tensor Scale** Figure 12 illustrates the impact of the input tensor's scale on the STE reconstruction error, plotted on a log-log scale. These plots show comparisons for a fixed block size of 16.

**Experiment 3: Softmax Error vs. Block Size** Similar to the first experiment, Figure 13 shows the reconstruction error for the Softmax approximation as a function of block size, comparing different data formats.

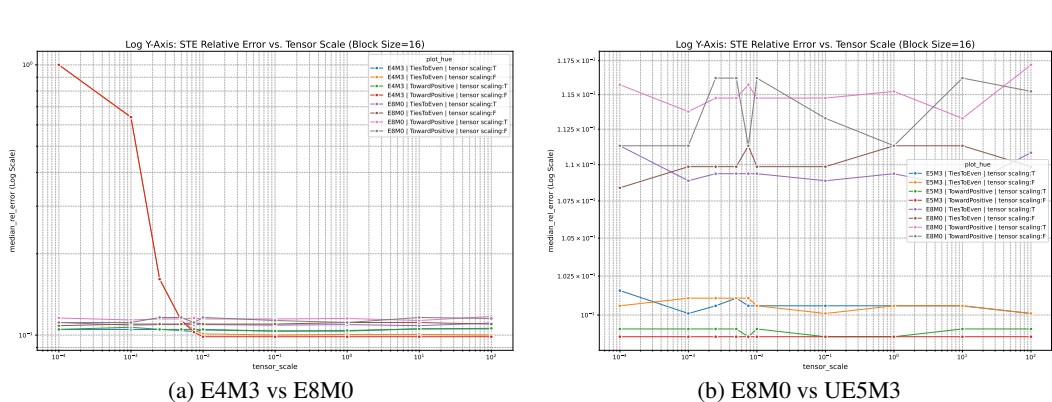

(a) E4M3 vs E8M0         (b) E8M0 vs UE5M3

Figure 12: Reconstruction error using STE as a function of tensor scale (Block Size = 16).

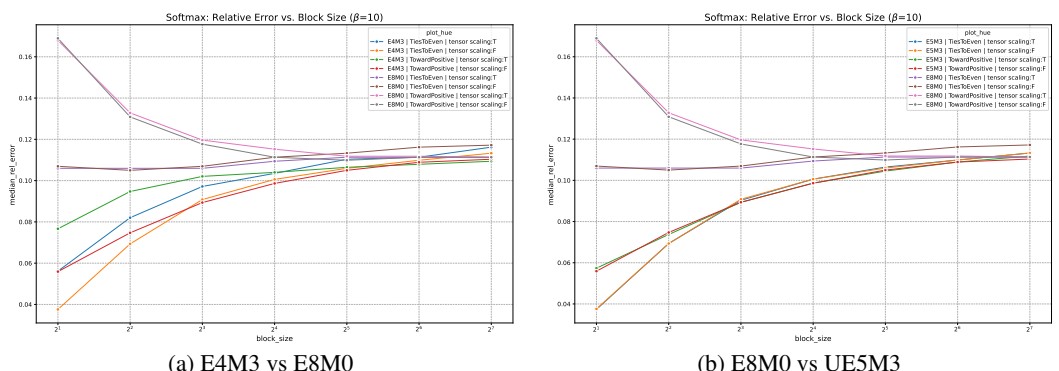

(a) E4M3 vs E8M0         (b) E8M0 vs UE5M3

Figure 13: Reconstruction error using Softmax approximation as a function of block size.

**Experiment 4: Softmax Error vs. Tensor Scale**   Figure 14 shows the effect of tensor scale on the Softmax approximation for a fixed block size of 16 and a $\beta$ value of 40.

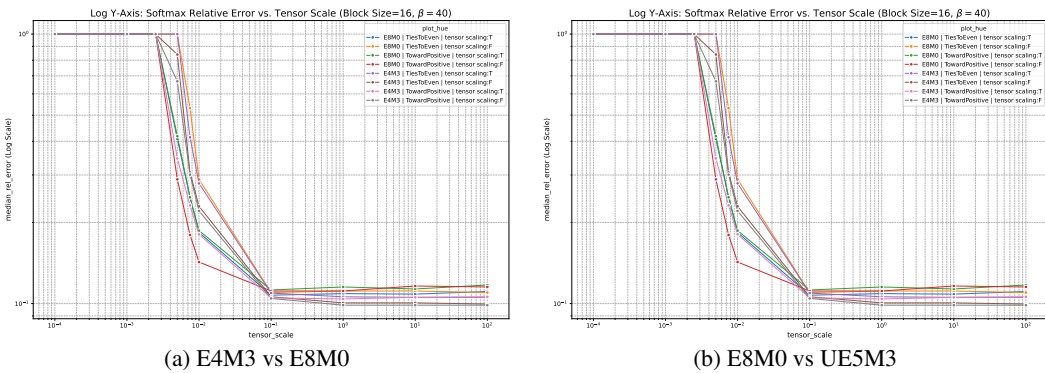

(a) E4M3 vs E8M0                                       (b) E8M0 vs UE5M3

Figure 14: Reconstruction error using Softmax approximation as a function of tensor scale (Block Size = 16, $\beta = 40$).

**Experiment 5: Softmax Sensitivity to $\beta$**   Finally, Figure 15 analyzes the sensitivity of the Softmax approximation to the inverse temperature parameter, $\beta$. The comparison highlights how tuning $\beta$ affects the reconstruction error for different formats.

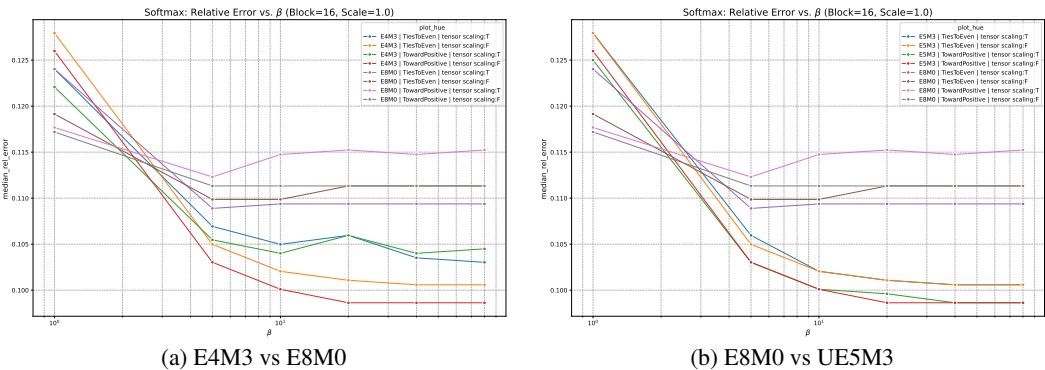

(a) E4M3 vs E8M0                                       (b) E8M0 vs UE5M3

Figure 15: Reconstruction error using Softmax approximation as a function of the $\beta$ parameter.

### A.13    THINGS WE TRIED BUT DIDN'T WORK

**Conditional Block-wise Scaling**   As scaling factors have limited range, we found in our initial experiments that the E4M3 format tends to stall during training, which is caused by underflow due to its limited range compared to E8M0. We propose a conditional scaling strategy, where the choice is determined by comparing the dynamic range of the data's scales, $\mathrm{DR}_{\mathrm{data}} = \frac{g}{\tilde{g}}$, with the intrinsic dynamic range of the target scale format, $\mathrm{DR}_{\mathrm{format}} = \frac{\mathrm{E4M3}_{\mathrm{max}}}{\mathrm{E4M3}_{\mathrm{min}}}$. Here, $g = \max_p\{m_p\}$ and $\tilde{g} = \min_p\{m_p\}$.

**Case 1: $\mathrm{DR}_{\mathrm{data}} \leq \mathrm{DR}_{\mathrm{format}}$ (Ideal Multiplicative Scaling)**   If the data's dynamic range fits within the format's range, we can compute a single constant $C$ using the geometric mean to center the scales within the target range:

$$C = \frac{\sqrt{\mathrm{E4M3}_{\mathrm{max}} \cdot \tilde{g} \cdot \mathrm{E4M3}_{\mathrm{min}} \cdot g}}{\mathrm{FP4}_{\mathrm{max}}}$$

The full forward pass for an element, including the final de-normalization, is:

$$h_{ij}(\mathbf{X}) = \frac{C}{q(C \cdot s_p)} Q\left(\frac{q(C \cdot s_p)}{C} \cdot \mathbf{X}_{ij}\right)$$

where $q(\cdot)$ is the quantization function for the scales (e.g., rounding to the nearest E4M3 value).

**Case 2: $\mathbf{DR_{data}} > \mathbf{DR_{format}}$ (Affine Mapping fallback)** If the scaled dynamic range is too wide, we resort to using an affine transformation to map $s_p \in \left[\frac{\mathrm{FP4_{max}}}{g}, \frac{\mathrm{FP4_{max}}}{\tilde{g}}\right]$ to the range $[\mathrm{E4M3_{min}}, \mathrm{E4M3_{max}}]$. The affine parameters are:

$$a = \frac{\mathrm{E4M3_{max}} - \mathrm{E4M3_{min}}}{\frac{\mathrm{FP4_{max}}}{\tilde{g}} - \frac{\mathrm{FP4_{max}}}{g}}, \quad b = \mathrm{E4M3_{max}} - a \cdot \frac{\mathrm{FP4_{max}}}{\tilde{g}}$$

The scale to be quantized is $\tilde{s}_p = a \cdot s_p + b$. We can then combine this with tensor scaling to achieve a reasonable quantisation:

$$h_{ij}(\mathbf{X}) = \frac{g \cdot \mathrm{E4M3_{max}}}{\mathrm{FP4_{max}} \cdot q(\tilde{s}_p)} Q\left(q(\tilde{s}_p) \cdot \frac{\mathbf{X}_{ij} \cdot \mathrm{FP4_{max}}}{g \cdot \mathrm{E4M3_{max}}}\right)$$

In the above setting, we're mapping the scale $s_p$ to the full range of E4M3, however due to the affine mapping we may lose precision for cases when $m_p \ll g$, since the term $\frac{\mathbf{X}_{ij} \cdot \mathrm{FP4_{max}}}{g \cdot \mathrm{E4M3_{max}}}$ will not have the full $\mathrm{FP4_{max}}$ range. We motivate this trade-off with the observation that NVFP4 has a block-size of 16, implying that having a well-represented scale outweighs the block accuracy.

When we tested the above on CIFAR10 as a unit test for E4M3 we couldn't get anywhere near convergence.

**Sigmoid approximation** Let $\mathcal{V} = \{v_1, \ldots, v_n\}$ denote FP4 (E2M1) levels. Define intervals $I_i = (v_i, v_{i+1}]$, $i = 1, \ldots, n-1$, with

$$c_i = \frac{v_i + v_{i+1}}{2}, \quad \Delta_i = v_{i+1} - v_i, \quad \gamma_i = \frac{12}{\Delta_i}.$$

For $x \in I_i$, let

$$z_i(x) = \frac{(x - c_i)\gamma_i}{T}, \quad w(x) = \sigma(z_i(x)) = \frac{1}{1 + e^{-z_i(x)}}.$$

**Proposition 3** (Smooth Quantization Properties). *Let $Q(x)$ be defined as above. Then:*

1. *The forward mapping $Q(x) = v_i + w(x)\Delta_i$ is a smooth interpolation between $v_i$ and $v_{i+1}$ using a sigmoid.*

2. *Its derivative is*

$$Q'(x) = \Delta_i \cdot \sigma(z_i)(1 - \sigma(z_i)) \cdot \frac{\gamma_i}{T} = \frac{12}{T}\sigma(z_i)(1 - \sigma(z_i)).$$

3. *In the limit $T \to 0$, $Q(x)$ converges to the standard ties-to-even quantization:*

$$\lim_{T \to 0} Q(x) = \begin{cases} v_i, & x \le c_i \\ v_{i+1}, & x > c_i \end{cases}.$$

*Proof.* The forward mapping is linear in $v_i$ and $v_{i+1}$ with a weight $w(x) \in (0, 1)$ from the sigmoid, so it is smooth and bounded by $v_i$ and $v_{i+1}$.

For the derivative:

$$Q'(x) = \frac{d}{dx}\left(v_i + w(x)\Delta_i\right) = \Delta_i \frac{dw}{dx} = \Delta_i \frac{dw}{dz_i}\frac{dz_i}{dx}.$$

Since $w = \sigma(z_i)$, we have $\frac{dw}{dz_i} = \sigma(z_i)(1 - \sigma(z_i))$, and $dz_i/dx = \gamma_i/T$, giving

$$Q'(x) = \Delta_i \cdot \sigma(z_i)(1 - \sigma(z_i)) \cdot \frac{\gamma_i}{T} = \frac{12}{T}\sigma(z_i)(1 - \sigma(z_i)).$$

Finally, as $T \to 0$, the sigmoid becomes a step function at $c_i$:

$$\sigma\left(\frac{(x - c_i)\gamma_i}{T}\right) \to \begin{cases} 0, & x < c_i \\ 1, & x > c_i \end{cases},$$

so $Q(x)$ reduces to ties-to-even quantization:

$$Q(x) \rightarrow \begin{cases} v_i, & x \leq c_i \\ v_{i+1}, & x > c_i \end{cases}.$$

$\square$

We tried this gradient adjustment, we expected it would provide a significant performance benefit, however this was not the case in early experiments (MNIST, gaussian regression, CIFAR10, llama 9M). Hence on lower level implementations, the additional complexity is not justified. The additional complexity is $\mathcal{O}(np \log k)$, with $\mathcal{O}(n)$ extra memory. $p$ denotes the number of polynomials used to evaluate the exponential function used in the sigmoid.