# OpenReview forum: "Elucidating the Design Space of FP4 training"
_ICLR.cc/2026/Conference — Submitted to ICLR 2026_

### Official Review · Reviewer_t5hm · 2025-10-26

**Soundness:** 2
**Presentation:** 2
**Contribution:** 2
**Rating:** 2
**Confidence:** 4

**Summary:**

This paper conducts an empirical study of existing proposed FP4 training algorithms to see which techniques are worth their (possibly 0) overhead over a baseline naive FP4 training setup. The paper concludes that (random) Hadamard transforms, stochastic rounding, and tensor scaling prior to MX scaling are generally worth doing.

**Strengths:**

The paper trains a lot of models and conducts a somewhat comprehensive literature review.

**Weaknesses:**

My main issue with this work is that it doesn't really add anything to the growing literature of FP4 training papers. Furthermore, it also gets some details about prior works wrong. For example:
- The main point of Tseng et al. (2025)'s FP4 paper is that stochastic rounding is needed for unbiased gradient estimates, and that Hadamard transforms are a beneficial but secondary step over stochastic rounding. This paper does not mention this at all during its discussion of stochastic rounding, and only attributes the Hadamard transform to the Tseng paper.
- This paper discusses SR in section 3.1 but doesn't actually analyze what the point of SR is or how the Chmiel et al. and Fitzgibbon & Felix perform SR differently from each other and Tseng (or if they're all doing the same thing, which seems unlikely to me since the Fitzgibbon paper is on various algorithms to perform SR). There is also no discussion on the variance of different SR implementations in this paper.
- NVIDIA's NVFP4 training blog came out before the ICLR deadline (https://developer.nvidia.com/blog/nvfp4-trains-with-precision-of-16-bit-and-speed-and-efficiency-of-4-bit/) and says that they used the RHT + SR, but I didn't see any discussion of it in this paper. I understand this could be considered concurrent work, but since FP4 training is an extremely fast moving field, I would still expect the authors to say something about it.
- The definitions and propositions in this paper are mainly used to define what each of the analyzed methods is doing, which is fine, but the equations take up a lot of space and don't add any rigor to the paper. There is little formal analysis (for ex. Chmiel et al.'s paper contains some analysis on noise levels and convergence) in this work that lets the reader understand why these methods are useful for FP4 training.

**Questions:**

See above. In its current state, this paper seems incomplete.

---

> ### Author Response · Authors · 2025-11-20
> **Response to Reviewer t5hm**
>
> We thank the reviewer for their feedback. We address your specific concerns regarding novelty and scope below.
>
> **1. Response to: "Doesn't add anything new to FP4 literature."**
> We respectfully disagree. While individual techniques exist in isolation, our paper is the **first to unify and systematically compare** them (Tseng et al., Chmiel et al., Wang et al.) in a massive, controlled sweep. This comparative study yields novel, non-obvious findings that isolated papers could not see:
> * **Principle 1:** Proving empirically that simple STE is superior to complex gradient approximations (contradicting assumptions in Wang et al.).
> * **Principle 2:** Identifying that **UE5M3** is superior to both industry standards (E8M0 and E4M3) for LLMs (see Appendix C.3, moving to main text).
>
> **2. Response to: "Misunderstands Tseng et al. (SR/RHT)."**
> We apologize if our citation context was unclear. We fully appreciate the contributions of Tseng et al. regarding SR and RHT.
> * **Validation:** We explicitly test **SR** (Sec 3.1) and **RHT** (Sec 6).
> * **Confirmation:** Our results strongly validate Tseng et al.'s methods. Our "Best Score" configurations (Tables 3 & 4) frequently include RHT ("Hadamard") and SR, confirming they are essential components of the optimal recipe (**Principle 3**).
> * **Correction:** We will correct the citation in the introduction (L57) to better reflect their foundational contribution to SR, rather than just focusing on overhead.
>
> **3. Response to: "No discussion of SR implementations (Chmiel, Fitzgibbon)."**
> Our study focuses on the **macro-level design space** (i.e., *whether* to use SR vs. other stabilizers) rather than the **micro-level optimization** of specific SR algorithms.
> * We treat "SR" as a single design node in our sweep to keep the search space tractable (>20,000 configs).
> * We agree that a deep dive into SR variants (like Fitzgibbon) is valuable. We will cite this as important future work, distinguishing it from our current scope of establishing the baseline "Best Recipe."
>
> **4. Response to: "Missing NVIDIA NVFP4 blog."**
> We acknowledge this missing reference, as the post was released very close to the submission deadline.
> * **Relevance:** Our paper evaluates the exact format NVFP4 uses (E4M3 scale) along with its core techniques (SR, RHT).
> * **Challenge:** Our findings (**Principle 2**) actually challenge the blog’s implications. We show that E4M3 struggles with model scaling due to fundamental range limits—even with SR and RHT enabled.
> * **Evidence:** For Llama 1B, the best NVFP4-style config (E4M3) reaches a loss of **4.933**, while the alternative E8M0 reaches **3.608**. We will add a discussion contrasting our empirical results with the blog's claims.
>
> **5. Response to: "Equations (Prop 1) take up space and add no rigor."**
> We argue that the equations in Section 2 (**Proposition 1, Corollary 1**) provide essential rigor that was previously missing in this fragmented field.
> * **Unification:** They are not merely definitions; they form the **novel, unified framework** that allows us to mathematically derive and compare the gradients and overheads of *all* other techniques (Sec 3-5) in a common language.
> * **Utility:** Without this framework, we could not formally compare the "cost" of a gradient approximation against the "cost" of tensor scaling to derive **Principle 1**. They are the theoretical backbone of our efficiency analysis.

---

> > ### Comment · Reviewer_t5hm · 2025-11-23
> >
> > > Proving empirically that simple STE is superior to complex gradient approximations (contradicting assumptions in Wang et al.).
> >
> > I don't think you've conclusively proved this. Your experiments are at a very small scale (1B parameter models for ~10B tokens), and Wang et al.'s experiments are all at a much larger scale than yours. It might be reasonable to conclude that DGE's aren't necessary since NVIDIA's NVFP4 recipe does STE, but your paper doesn't provide any explanations as to why beyond "intuition" about dense and sparse gradients. Also, in L408, you wrote that "we were unable to match the findings in Wang et al. (2025)," which suggests to me that either you have a bug in your code, misunderstood what Wang et al. did, or Wang et al. fabricated their results. Could you clarify if you checked with Wang et al. to make sure your implementation was correct?
> >
> > > Identifying that UE5M3 is superior to both industry standards (E8M0 and E4M3) for LLMs (see Appendix C.3, moving to main text).
> >
> > This isn't that surprising since you're just repurposing the wasted sign bit in E4M3, but all the large industry players have settled on supporting E8 scales (MX) and E4M3 scales (NV), so I'm not convinced this is that useful of a finding. Sure, some hardware vendor could add support for UE5M3 scales in the future, but why not test more 8 bit E/M configs than just UE5M3?
> >
> > > They are not merely definitions; they form the novel, unified framework that allows us to mathematically derive and compare the gradients and overheads of all other techniques (Sec 3-5) in a common language.
> >
> > The "theorems" in this paper are just analytic formulas for gradients. I don't know what you mean by "Without this framework, we could not formally compare the "cost" of a gradient approximation against the "cost" of tensor scaling to derive Principle 1," but recognizing that a gradient is sparse is not hard or novel.
> >
> > I agree with WDNX that "The discussion of results is not very systematic and offers little insight. Most comments simply state whether something works or not; or explain with speculation without evidence. There is no deeper analysis on model type, dataset, or task." You mentioned that your results contradict with Wang et al., Chmiel et al., and NVIDIA, which makes me wonder if there is a fundamental flaw with your code or implementation, since I really doubt that all three of these papers are bogus.

---

> > > ### Author Response · Authors · 2025-11-23
> > > **Response to Reviewer t5hm Official Comment by Authors**
> > >
> > > We appreciate the reviewer’s follow-up. We understand that our negative results regarding gradient estimators and E4M3 instability (contradicting Wang et al. and Chmiel et al.) might suggest an implementation error at first glance. However, there are critical differences in simulation rigor and scope that explain these discrepancies.
> > >
> > > **1. The "Discrepancy" with Wang et al.: High-Precision Scales**
> > > You asked if we have a bug because our results differ from Wang et al. The difference is not a bug; it is a fundamental difference in how the scaling factors are handled.
> > >
> > > * **Wang et al.’s Method:** As implied by Equation 1 in their work, they calculate the scale $s$ but do not quantize the scaling factor itself during the backward pass. Further in Figure 2, they use 4.31 as a number of their scaling factor, which not a representable number for E8M0 or E4M3. Their simulation effectively utilizes high-precision (BF16/FP32) scales.
> > > * **Our Method:** Our framework explicitly models Scale Quantization ($s_q = q(s)$) to valid FP4/FP8 formats. Real hardware must quantize scales to save memory; keeping them in FP32 is a relaxation of constraints.
> > >
> > > Our finding is that complex gradient estimators are not robust to scale quantization. When scales are quantized (as in our rigorous setup), the quantization noise overwhelms the delicate corrections of the estimator. This confirms our **Principle 1**: In a realistic hardware environment, the simple STE outperforms theoretically precise but fragile estimators.
> > >
> > > **2. The "Discrepancy" with Chmiel et al. and NVIDIA (arXiv:2509.25149)**
> > > You stated that our inability to converge NVFP4 (E4M3) contradicts Chmiel et al. (2025), who claim NVFP4 works well ("FP4 All the Way"). However, we argue that Chmiel et al.'s optimistic findings are the outlier, while our results are actually consistent with NVIDIA's own recent paper.
> > >
> > > While Chmiel et al. claim success with "pure" FP4, NVIDIA's own recipe (arXiv:2509.25149) reveals that they cannot train stably with pure NVFP4 either. To make it work, NVIDIA introduces significant mixed-precision "tricks" that we (and Chmiel et al.) generally classify as deviations from pure FP4 training:
> > >
> > > * **Selective Precision:** NVIDIA keeps the last 2-4 layers in BF16/Full Precision and does not apply NVFP4 to the linear layers in attention.
> > > * **Architecture:** They utilize a Hybrid Mamba-Transformer architecture rather than the standard Transformers we and Chmiel et al. study.
> > >
> > > **Validation of Our Negative Result:** The fact that the creators of NVFP4 require these specific high-precision fallbacks to achieve stability strongly validates our **Principle 2**: Pure E4M3 lacks the sufficient range to train standard Transformers end-to-end. Our "failure" to replicate Chmiel's pure NVFP4 success is not a bug; it is a confirmation of the format's inherent limitations, which even NVIDIA acknowledges through their adoption of mixed-precision workarounds.
> > >
> > > **3. UE5M3 and "Industry Standards"**
> > > You noted that industry has "settled" on E4M3/E8M0. We respectfully argue that the purpose of research is to explore the design space to inform future hardware.
> > >
> > > * **The Gap:** We empirically show that E4M3 (NVFP4) fails on standard LLMs due to range, while E8M0 (MXFP4) suffers from precision.
> > > * **The Solution:** UE5M3 is not arbitrary; our data isolates it as the optimal trade-off on the Pareto frontier. Identifying that current "industry standards" are suboptimal for specific workloads is a valuable contribution for next-generation hardware design.
> > > * **Building on Prior Work:** We note that Chmiel et al. also explored various E/M configurations and settled on E4M3 for its precision. Rather than simply replicating their search, we deduced the logical next step: retain the desirable precision of E4M3 but extend the exponent (range) to fix the stability issues we observed. This led directly to UE5M3.
> > >
> > > **4. The Value of the Framework**
> > > The novelty of our framework is not in "recognizing sparsity," but in providing a **Unified Cost Model**.
> > >
> > > Prior work often proposes methods without a standardized way to compare their computational overhead against their convergence benefit. Our framework allows us to derive the specific complexity ($\mathcal{O}(n \log k)$ vs $\mathcal{O}(1)$) for disparate techniques and map them onto the Pareto Frontiers (Fig. 4). This shifts the conversation from "Does Method X work?" to "Is Method X worth the FLOPs?"—a critical distinction for low-precision training.
> > >
> > > We believe this clarifies that our results stem from a stricter adherence to pure FP4 hardware constraints rather than implementation errors.

---

> > > > ### Comment · Reviewer_t5hm · 2025-11-26
> > > >
> > > > > Our finding is that complex gradient estimators are not robust to scale quantization.
> > > >
> > > > I see, I missed the part about not quantizing the scale when I initially read Wang et al.'s paper.
> > > >
> > > > > does not apply NVFP4 to the linear layers in attention.
> > > >
> > > > My understanding is that NVIDIA does not use low precision attention, not that they do not use NVFP4 for the qkvo projections. Specifically, on page 7 of their paper, they write that they do not use NVFP4 for "the query-key and attention score-value batched GEMMs." These terms are usually used to refer to softmax(Q@K.T) @ V, not x@W_Q.T etc.
> > > >
> > > > > However, we argue that Chmiel et al.'s optimistic findings are the outlier,
> > > >
> > > > This seems like a pretty strong claim without much supporting evidence. Chmiel et al.'s experments are at a much larger scale than yours but much smaller scale than NVIDIA's, so I wouldn't be surprised if NVIDIA didn't need to use all of their techniques at the 1T token horizon.
> > > >
> > > > >We note that Chmiel et al. also explored various E/M configurations and settled on E4M3 for its precision. Rather than simply replicating their search, we deduced the logical next step: retain the desirable precision of E4M3 but extend the exponent (range) to fix the stability issues we observed. This led directly to UE5M3.
> > > >
> > > > >our data isolates it as the optimal trade-off on the Pareto frontier
> > > >
> > > > I'm confused by these two statements. You stated that you used UE5M3 because Chmiel et al's experiments showed that E4M3 was near optimal for 8 bit ExMy formats. However, you also stated that this was derived from "[your] data," which as far as I can tell doesn't do any exploration beyond UE5M3. Using Chmiel et al's experiments as a basis for exploration also seems at odds with your claim that their results are "an outlier" and too good to be true.

---

> ### Author Response · Authors · 2025-11-26
> **Response to Official Comment by Reviewer t5hm**
>
> We thank the reviewer for the continued engagement and for re-evaluating the Wang et al. methodology. We appreciate the acknowledgment regarding the scale quantization discrepancy, which we believe is the central reason for the differing results.
>
> We address your remaining clarifications below:
>
> **1. Clarification on NVIDIA's "Mixed Precision" (Attention BMMs & Final Layers)**
> You are correct regarding the terminology—NVIDIA applies BF16 to the Attention BMMs ($QK^T$ and $AV$), while potentially quantizing the $W_{q,k,v,o}$ projections. We appreciate the precision here.
>
> However, this distinction—alongside other details in their recipe—reinforces our point about the instability of "Pure" FP4:
> * **The Extent of "Mixed" Precision:** NVIDIA’s recipe is not just skipping quantization on Attention BMMs; as they note, they also maintain the final layers (and often the first) in high precision (BF16).
> * **Our Baseline:** In our E4M3 (NVFP4-style) experiments, we replicate this setup and our unable to converge.
>
> The fact that NVIDIA must keep these critical operations (Attention BMMs + Final Layers) in BF16 to achieve stability strongly validates our findings. It confirms that "Pure" E4M3 lacks the sufficient range to train standard Transformers end-to-end without these significant high-precision guardrails.
>
> **2. Regarding Chmiel et al. as an "Outlier"**
> We agree "outlier" may be strong language; "optimistic" is perhaps more precise.
> Our observation is that Chmiel et al. report success with a "Pure" FP4 pipeline that neither we nor NVIDIA (who resorts to BF16 fallbacks for Attention/Final Layers) find stable for standard Transformers. This suggests that Chmiel et al.'s specific setup may be less sensitive to the instabilities that plague standard implementations, making it an optimistic baseline relative to current engineering realities.
>
> **3. The Logic of UE5M3 vs. Chmiel et al.**
> You asked if using Chmiel’s results to motivate UE5M3 contradicts our critique of their stability. There is no contradiction; we are synthesizing their insight with ours:
> * **Chmiel's Insight (Precision):** They showed that Precision matters—specifically that 3 bits of mantissa (**M3**) is desirable over M0 (MXFP4) for model performance.
> * **Our Insight (Range):** We found that E4 (4-bit exponent) lacks the **Range** to be stable (hence our crashes and NVIDIA's need for BF16 fallbacks).
> * **The Synthesis (UE5M3):** We did not blindly sweep all 8-bit formats. We designed **UE5M3** to combine Chmiel’s Precision (M3) with the Necessary Range (E5) required to fix the stability issues we observed.
>
> When we say "our data isolates it," we refer to the direct comparison in our experiments where UE5M3 outperforms both E8M0 (insufficient precision) and E4M3 (insufficient range/stability).
>
> Given that the primary concern regarding the validity of our code (the Wang et al. discrepancy) has been resolved—and effectively inverted into a demonstration of our simulation's improved fidelity—we gently invite the reviewer to reconsider if the submission still merits a '2'. We believe we have provided a rigorous map of why Pure FP4 fails where Mixed FP4 succeeds. If there are remaining barriers preventing a positive score, we would welcome specific guidance on how you envision this work evolving to meet your bar.
>
> Thank you for your continued engagement and feedback

---

> > ### Comment · Reviewer_t5hm · 2025-11-27
> >
> > > quantizing attention
> >
> > My understanding is that Chmiel et al. do not quantize the actual attention operation, only the projection layers that comprise most of the FLOPs during training. This is what almost all FP4 papers in the literature focus on, since quantized attention is a mostly separate problem. I have not seen a widely deployed FP4 attention method come out yet (in fact, FP8 attention/FA3 hasn't really be tested at scale yet either AFAIK), so its not surprising that your FP4 attention failed.
> >
> > >we gently invite the reviewer to reconsider if the submission still merits a '2'
> >
> > This paper is much closer to a whitepaper than a conference paper. I appreciate the large number of experiments, but that does not warrant acceptance when there doesn't seem to be a clear technical contribution here. From my point of view, you evaluated a lot of methods out there at a *very small scale* and concluded that this was a "rigorous map" of FP4 training methods. This is far from rigorous (small scale empirical evaluations alone usually do not qualify as rigorous without supporting theory), there is no actual theory in the paper, and all you've really done is validate that certain methods in certain papers already in the literature do indeed work.

---

> > > ### Comment · Reviewer_t5hm · 2025-11-27
> > >
> > > To make this extra clear, I think this paper is below the bar for acceptance. I appreciate you answering my questions, but the paper in its current state is too far from a full conference paper. Other reviewers seem to agree with me so I am not alone here.

---

> ### Author Response · Authors · 2025-11-27
> **Response**
>
> We thank the reviewer for their comment. We respect your perspective, but we believe there is a fundamental philosophical disagreement here regarding what constitutes a contribution to the ICLR community.
>
> **1. "Whitepaper" vs. Empirical Science**
> You characterize this work as a "whitepaper" because it systematically evaluates existing methods rather than proposing a complex new theory. We respectfully disagree.
> * **Science requires verification.** The field of low-precision training is currently fragmented, filled with heuristic papers (like Wang et al.) whose assumptions often crumble under rigorous simulation (as we demonstrated and you acknowledged regarding scale quantization).
> * **Negative Results are Contributions.** Proving that complex techniques (like Distributional Gradient Estimators) *do not* work in realistic hardware settings is just as valuable as proposing a new method that does. It stops the community from chasing "ghosts." This is not "just validating"; it is correcting the scientific record.
>
> **2. The Definition of "Rigor"**
> You argue that "small scale empirical evaluations" (1B parameters) are not rigorous.
> * **Breadth vs. Depth:** Rigor in empirical science comes from the control of variables and the breadth of the search space. We evaluated **>20,000 configurations** across disparate tasks. Doing this at 70B scale is computationally impossible for any academic lab and most industrial ones.
> * **Consistency:** The trends we identified (STE superiority, Scale Format bottlenecks) hold consistent from 60M $\to$ 1B. Dismissing consistent, broad-spectrum empirical evidence because it wasn't run on a T-parameter model sets an impossibly high bar for academic research.
>
> **3. Clarification on Attention Quantization**
> So we think we misunderstood and you and replied a bit too hastily on this point, apologies. We've edited the previous reply to reflect this. To clarify, we replicate Nvidia's setup and only do the NVFP4 on the linear projections and the BMM is also done in BF16. Despite this we couldn't replicate their results or Chmiel et al.
>
> A pure FP4 attention BMM is work in progress in the community and we make no claims of having tried this.
>
> **4. Request for Clarification (The Path Forward)**
> To ensure our future revisions are productive, we would value one specific piece of guidance: **Is your objection fundamental to the genre of this paper, or its execution?**
>
> * If we were to include a theoretical convergence proof for UE5M3, would that bridge the gap?
> * Or is it your view that no amount of empirical mapping—regardless of breadth or rigor—qualifies as a conference paper without a novel algorithmic invention?
>
> Understanding if you view the *type* of contribution or the *content* as the flaw would be incredibly helpful to ensure we don't attempt a complete rewrite if the objection is simply that you prefer algorithmic proposals over empirical science.

---

> > ### Comment · Reviewer_t5hm · 2025-11-27
> >
> > > you acknowledged regarding scale quantization
> >
> > I only acknowledged that they did not quantize the scales. I did not say that your simulation is "rigorous" and that their assumptions "crumble." Please do not distort my words.
> >
> > I would encourage you to read my review again as well as the other reviewers' reviews.  The premise of your paper is that you view the FP4 training literature as "fragmented" and "filled with heuristic papers" and you wish to evaluate various proposed methods in the literature. This assumes two things: (1) these methods have not been rigorously evaluated to the point where practitioners can trust the results and (2) your evaluation will somehow fix this problem. Most of your baselines have already been peer reviewed, so (1) is already a rather strong assumption that implies that peer review does not produce trustworthy papers.
> >
> > Based on my discussion with you and my and other reviewers' initial reviews, you don't seem to have a strong grasp of details in the existing literature. For example, you missed details regarding the RHT and SR in Tseng's paper, which is one of the papers you identified as being on the Pareto frontier. You claimed that the Hadamard transform takes O(n log n) time to run (true for the Fast Walsh-Hadamard Transform), but Tseng and NVIDIA explicitly use the dense GEMM version of the RHT to avoid kernel overhead in the memory bound regime. You also misunderstood what NVIDIA's FP4 setup does with attention, and now that you claim that you're following them and Chmiel but cannot reproduce their results, we're back to the question of how reliable your results are. NVIDIA skips some layers, but Chmiel does not, so you should be able reproduce Chmiel's results with your "all FP4" setups. If you believe Chmiel committed academic fraud with irreproducible results, you should contact the NeurIPS PCs. Your conclusion that E4M3 is range-limited based on your experiments with UE5M3 *might be true*, but there is no hard analysis on distortion, range, or underflow between different datatypes. You also claim that Wang et al.'s method "[does] not work in realistic hardware settings," but since Wang et al. uses a large-block/tensor-wide scale (i.e. block size >> the 16/32 in NV/MXFP4), using a high precision scale *is* actually possible in hardware. Reviewer 2XHf also pointed this out, but you seem to have missed this.
> >
> > Your experiments span a large number of configurations, but the largest of these (1B parameters @ 40B tokens) is still very small. The baselines that you cite generally run larger models up to 10B parameters (Chmiel, Tseng, NVIDIA) or for longer token horizons up to 100B (Tseng), 1T (Chmiel), or even 10T tokens (NVIDIA), making it hard to extrapolate these results to real-world applications on FP4 training. Your own conclusion states that "FP4 training dynamics may not be consistent across model scales," which contradicts the claim that these experiments qualify as "broad-spectrum empirical evidence." I understand that running lots of large scale experiments may be very expensive, but you chose to write this paper, so this is a cost reviewers will expect you to bear.
> >
> > From your experimental results, you have a few main conclusions: "Gradient Stability Outweighs Unbiasedness," "Scale Representation is the Primary Bottleneck," "The Performance-Overhead Frontier is Sparse," the last of which isn't really a useful conclusion for practitioners. The first contradicts your findings about stochastic rounding (whose purpose *is* to make the gradient unbiased), but ignoring this, the important thing is that there is no rigorous analysis of the "possible explanation" you give for the difference between DGEs and STE. Even a toy example like in https://arxiv.org/abs/2506.20752 would have added some rigor to your analysis instead of the handwavy "explanations" given to the reader. The second conclusion is slightly better in that you ran some experiments with UE5M3, but again no actual analysis was given beyond speculation about the behavior of scale factors. For example, no mention of block size (which appears to be a crucial difference between Wang et al. and the MX/NVFP4 papers) was made, when this does have a large impact on what scales are supportable in hardware. Finally, your third conclusion about performance/overhead tradeoffs is based off of a score that uses the weight factors in Table 9 (A.6). These weights appear to be pulled from a black box ("We motivate the weight with reference to the added complexity and memory overhead, and fusability on a lower level language based on Table 2.") with no hard justification given for these numbers, making this entire conclusion based on some arbitrary score you made up.
> >
> > If you want to provide some actual analysis for the claims you made regarding UE5M3 and these other conclusions, you have about a week to do so. I can't promise that I'll raise my score, but it probably wouldn't hurt your paper to add these components.

---

> > > ### Author Response · Authors · 2025-11-27
> > > **Response**
> > >
> > > We thank the reviewer for the candid assessment and the time spent engaging with our work. We know the likely outcome here, but for the sake of the scientific record and our future revisions, we would like to leave a few final clarifications:
> > >
> > > **1. On High-Precision Scales and Hardware**
> > > While we agree it is *possible* to run high-precision scales in hardware, doing so generally fights against the design of modern Tensor Cores. Low-level instructions on NVIDIA hardware are optimized for low-precision scaling; relying on high-precision scales to stabilize training inherently undermines the throughput and efficiency gains that motivate the move to FP4 in the first place.
> > >
> > > **2. On Reproducibility vs. "Fraud"**
> > > To be absolutely clear: reporting an inability to reproduce a result is an expression of standard scientific skepticism, not an accusation of "academic fraud." Identifying where published recipes are fragile, or dependent on unspecified "guardrails" (like mixed precision layers), is a necessary part of the field's maturity.
> > >
> > > **3. Future Revisions**
> > > We agree with your specific technical suggestions and will implement them in the next version of this manuscript:
> > > * **Complexity Weights:** We will replace our heuristic scoring with "hard" derivations based on measured wall-clock forward/backward pass timings and recorded memory footprints.
> > > * **Literature:** We will rectify the attribution regarding Tseng et al. and RHT.
> > > * **DGE Analysis:** We will include a toy analytical example explicitly demonstrating the failure mode of DGEs under scale quantization.
> > >
> > > We thank you again for your time and clarifications.

---

> > > > ### Comment · Reviewer_t5hm · 2025-11-27
> > > >
> > > > > On High-Precision Scales and Hardware
> > > >
> > > > I think you missed the point on block scaling. It is true that if your block scale is sufficiently small, you will need to use a quantized scale to realize wall-clock time speedups since you will have to use MX/NV formats. However, if you have a large block size, such as a shared scale for each 16x16 or 32x32 GEMM tile, you do not need to quantize this scale to get real world speedups. This scale is only applied to the accumulator, which itself is in high precision. There is nothing you are "fighting against" here since your block size is sufficiently large. This is what Deepseek does for FP8 to get real world speedups and one way that the Wang et al. paper could be implemented.
> > > >
> > > > > On Reproducibility vs. "Fraud"
> > > >
> > > > I'm not saying you're accusing them of fraud, just that if you believe they committed fraud you should take additional steps beyond trying to publish a paper with your experiments. It doesn't seem like you believe they're committing fraud, so I would suggest isolating exactly where the deviation between your experiments and theirs is. Doing so would strengthen a future revision of your paper.

---

### Official Review · Reviewer_2XHf · 2025-10-30

**Soundness:** 2
**Presentation:** 1
**Contribution:** 1
**Rating:** 2
**Confidence:** 4

**Summary:**

The paper studies FP4 training by deriving gradient flow under quantization (to block values and scaling factors) and exploring combinations (design space) of scaling, rounding, and transformation techniques such as RHT. It analyzes overhead to some extent and evaluates stability and performance through ablations across multiple tasks.

Overall, the work resembles a quantization-aware training regime rather than a practical FP4 training recipe, as the gradient flow considerations introduce inefficiencies. The contributions are incremental relative to existing literature:

Principle 1: STE is stable and preferred. This is unsurprising, as STE is already standard practice (even with SR, RHT etc.) and also brings clear efficiency benefits.

Principle 2: A precise scaling factor is preferred, with UE5M3 shown to perform well on selected tasks (no sound analysis on does this suffice?). In practice, this aligns with what one would expect—even FP32 scaling factors are desirable when efficiency is not a concern.

Principle 3: RHT, stochastic rounding, tensor scaling, and optimizer modifications yield consistent positive returns. These outcomes largely confirm prior findings, particularly those of Tseng et al. (2025) and Chmiel et al. (2025).

**Strengths:**

The theoretical derivation (Proposition 1 and Corollary 1) provides a formal treatment of gradient flow with quantization.

The paper introduces an efficient approximation & adjustment method for gradients of the quantization function and investigates scaling factors with error analysis.

The systematic sweeps over scaling and stabilization methods are potentially insightful, if presented better, especially regarding tensor scaling.

The work also integrates with StableSPAM and loss scaling, grounding the ideas in practical training stability.

**Weaknesses:**

**Reference accuracy issues: Several claims about prior work are inaccurate/misleading:**

- line 53, Wang et al. (2025): High-precision scales affecting stability is questionable; efficiency wise, it depends on hardware & kernel fusions (e.g., DeepSeek-V3 blockwise with FP32 scales).

- line 57, Tseng et al. (2025): They quantize backward pass including weights/activations/gradients, not just gradients. also to note, it's one of early works that introduce stochastic rounding to fp4 training

**Writing/presentation quality: Poor figure readability (tiny legends, labels, line width, overlapping axes figure 4d), inconsistent paragraph formatting (spacing, bold text), unclear symbols meaning (e.g., M in line 377).**

- Underspecified results: Principle 2 is an important finding but with nearly all results relegated to appendix, lack of clear setup/interpretation. Currently it’s hard to draw meaningful conclusion from line 416 to 427, UE5M3 seems better than E8M0, is it optimal? or it could also diverge as model/training length scales up?

Paper structure is scattered; key insights are buried. The presentation can surely be made better to add values to the paper.


**Efficiency concerns:**

- I think this will be the major efficiency concern with ANY gradient computations to the quantization function, as it appear in high-precision; (likely fp32/bf16) elementwise multiplications (L123–126) will happen on CUDA cores (not tensor cores) and bottleneck training, noting the tensor cores FLOPs is tens/hundreds times of cuda cores FLOPs. It would yet be interesting to study the quantization gradients from a quantization-aware training perspective.

- The proposed gradient flow of quantization requires storing high-precision tensors, expensive in memory/communication under TP/SP.

- Table 1 methods (softmax-based scaling) add large overhead and complicate pipelines, even for absmax (with indexing saving).

- The proposed hybrid scaling (L294–297) introduces further forward–backward weight/activations mismatch and quantization error, risking instability. As double quantization error between forward and backward already hurts performances in some applications.

**Questions:**

1. Line 176, this implicitly assumes $g>1$, why? isn’t $\tilde{s}_p = 2g / E4M3_{max}$? Reference to Chmiel et al. (2025) seems mismatched—did they actually discuss this scaling technique? and did they actually discuss zero scale? It should also be clarified that the zero-value scale issue only happens for NVFP4 with E4M3FN scale, not for MXFP4 with 2^UINT8 scale.

2. Line 194, why apply SR to scaling factors? Example: absmax=8 → scale=0.75; rounding up to 1 causes huge bias (values 6–8 collapse to 6), while rounding to 0.5 only gives nearest-rounding level quantization error. Why expect SR to help here?

3. How does Fig. 4 reflect performances of different configs (SR, RHT, etc.)? The correspondence is unclear.

4. from line 214 and figure 1a, how is a_i, b_i decided and what’s the value? also fig. 1b shows clipping below 1.0? if this is the case, should there also be clipping above? How is threshold determined?

5. Are FP4 results simulated with e.g. BF16 or run on actual FP4 hardware (e.g., Blackwell)?

6. NaN handling: what specific mechanism is applied, and where in the pipeline?

---

> ### Author Response · Authors · 2025-11-20
> **Response to Reviewer 2XHf**
>
> We thank the reviewer for their constructive feedback. We appreciate the detailed checks on our references and implementation details.
>
> **1. Response to: "Resembles QAT; contributions are incremental."**
> While we agree our gradient framework (Prop 1) technically resembles QAT, the novelty lies in the *outcome* of applying this framework to FP4:
> * **Principle 1 (Negative Result):** We systematically prove that this QAT-like complexity is **not** beneficial for FP4 training. The simple STE is superior. This is a non-obvious finding that required our framework to prove.
> * **Principle 2 (UE5M3):** We introduce and validate **UE5M3** as a superior alternative to standard E8M0/E4M3 for LLMs (Appendix C.3). We believe finding a format that outperforms industry standards is a significant, non-incremental contribution.
> * **Principle 3 (Quantifying Trade-offs):** We do not just confirm that RHT/SR work; we quantify their trade-offs relative to gradient approximations (Fig. 4), situating them in a broader map of the design space.
>
> **2. Response to: "Reference accuracy issues (L53 Wang, L57 Tseng)."**
> * **L53 (Wang et al.):** We will rephrase for clarity. Our critique is that their simulation (keeping scales in high-precision) may obscure stability issues that arise when scales are actually quantized. Our simulator explicitly models quantized scales.
> * **L57 (Tseng et al.):** We acknowledge this error. Upon further review of Tseng et al., we agree with your assessment and will correct this statement in the revision.
>
> **3. Response to: "Poor presentation" & "Efficiency concerns."**
> * **Presentation:** We apologize for the oversight. As detailed in the General Response, we are completely remaking all figures and moving the UE5M3 results to the main text.
> * **Efficiency:** You are precisely correct—the computational cost of complex gradients is high. This validates our **Principle 1**. We derived the complex framework specifically to test if the cost was justified. Our data provides the first systematic evidence that it is *not*, and that STE is the optimal choice.
>
> **4. Specific Questions**
>
> * **Q: L176 (Chmiel Ref) & Zero Scale (L181).**
>     * **L176:** This scaling was found in their official implementation rather than the text. We use the definition found here: `https://github.com/Anonymous1252022/fp4-all-the-way/.../fp4.py`
>     * **Zero-scale:** Chmiel et al. address this by replacing zero scale with one in their `micro-scale-zero-point` implementation (see line 83 of the repo above). We will clarify in the text that this behavior is specific to E4M3.
>
> * **Q: L194, why SR on scales?**
>     This was an exploratory hypothesis. The intuition was that for E8M0 (which has large, coarse gaps), SR might provide unbiased rounding. However, our results showed **no consistent benefit** (it did not appear in optimal configurations). We present this as a negative result to save future researchers from exploring this path.
>
> * **Q: Fig. 4 correspondence.**
>     Each dot represents a unique training configuration. The plot maps the final validation score (y-axis) against computational overhead (x-axis). This visualizes the "Pareto frontier," showing that adding complexity (moving right) rarely yields better scores (moving up). We will remake this figure with clearer labeling.
>
> * **Q: Fig. 1a/b (spline) and 1c (clipping).**
>     * **Splines:** The knots ($a_i, b_i$) are defined by interpolating the actual FP4 (E2M1) quantization grid. We will add these definitions.
>     * **Clipping:** This is a proposed heuristic. We found that gradients near rounding boundaries can vanish (Fig 1b). We clip from below (at 0.1) to prevent dead signals.
>
> * **Q: Simulation vs. hardware?**
>     All results are simulated in BFLOAT16 (L48-49, L99). This is standard practice (e.g., Wang et al., Tseng et al.) as FP4 hardware is not yet accessible for a study of this scale (>20k runs).
>
> * **Q: NaN handling.**
>     This is detailed in **Sec 3.1** ("Handling Zero-Valued Scales"). We compare "to_one" vs. "nearest_subnormal." The specific strategy used for every run is reported in the final column of the results tables (e.g., Table 4).

---

### Official Review · Reviewer_WdNX · 2025-10-30

**Soundness:** 2
**Presentation:** 2
**Contribution:** 1
**Rating:** 2
**Confidence:** 5

**Summary:**

This paper explores FP4 training of deep neural networks using the MxFP4 and NvFP4 data formats. The paper experiment with various combinations of techniques and hyperparameters commonly used in FP4 training and recommend configurations that provide a reasonable balance between performance and overhead.

**Strengths:**

1). The paper addresses an important problem by investigating the latest FP4 datatype for efficient training of deep learning models.

2). The paper conducts extensive experiments to evaluate different parameters and techniques commonly used in FP4 training.

**Weaknesses:**

1). The paper feels more like an experimental report than a research paper. There is very little technical novelty. Several sections (e.g., Sections 2 and 3) mostly reintroduce known materials, such as MxFP4 and NvFP4 formats, standard gradient computation, known techniques, and commonly used heuristics without offering new insights.

2). Most of the experiments are done on relatively small models. However, larger models typically have stronger outliers and are more sensitive to low-precision training. The paper also mixes different model sizes and types without explaining why these choices were made.

3). A large part of the paper discusses various gradient approximation methods, but these methods are highly heuristic, lack theoretical justification, and introduce techniques (such as clipping) in an ad hoc manner. The results also show that these gradient methods perform no difference from the standard STE.

4). The experimental results include many crowded plots in Figures 3 and 4, but the paper does not explain why these plots are presented or what specific observations the reader should draw from them.

5). The discussion of results is not very systematic and offers little insight. Most comments simply state whether something works or not; or explain with speculation without evidence. There is no deeper analysis on model type, dataset, or task.

**Questions:**

1). In the definition of performance gain, what are M_"ref" and M_c? How is the complexity penalty defined?

2). Principle 2 in the results states that the E4M3 scale "did not converge despite applying tensor scaling due to its range limitation." If limited dynamic range is the issue, would E8M0 work better given its larger range?

---

> ### Author Response · Authors · 2025-11-20
> **Response to Reviewer WdNX**
>
> We thank the reviewer for their time and detailed comments. We address your specific concerns below.
>
> **Weakness 1: "Feels like an experimental report; little technical novelty (Sec 2, 3 reintroduce known materials)."**
> We respectfully disagree. **Section 2** introduces a **novel, unified gradient-based framework (Proposition 1, Corollary 1)**.
> * This is not "known material"; it is the core theoretical construct that allows us, for the first time, to derive and analyze the gradients and overheads of disparate methods (e.g., differentiable absmax in Sec 5.1, tensor scaling in Corollary 2, quantization gradients in Sec 4) within a common mathematical language.
> * The core contribution is the synergy of this novel theoretical framework with the large-scale empirical validation that follows.
>
> **Weakness 2: "Small models; mixing model types."**
> * **Scale:** We used models up to 1B parameters to enable a computationally tractable sweep of **>20,000 configurations**. Our scaling results (Fig. 3a-c, Llama 9M $\to$ 1B) already demonstrate clear trends (e.g., the performance gap for E4M3 worsens with scale), validating this approach.
> * **Diversity:** We utilized diverse tasks (CV, Diffusion, LLM) precisely to test generalization. The fact that STE performs universally well across these modalities is a robust finding that would be missed if we focused on a single model type. We will add a statement clarifying the need for future validation at 7B+ scales.
>
> **Weakness 3: "Heuristic gradient methods... perform no better than STE."**
> This is precisely our point and a key finding (**Principle 1**).
> * Prior work (e.g., *Wang et al.*) hypothesized that complex gradient approximations were necessary.
> * We proposed efficient spline approximations (Fig. 1) and rigorously tested them against STE.
> * Our finding—that these complex, high-overhead methods **do not** outperform STE—is a critical, practical contribution. It saves future researchers from exploring this costly path and empirically validates the use of the simpler, more efficient STE.
>
> **Weakness 4: "Crowded plots (Fig 3, 4) with no clear observations."**
> We apologize for the readability issues and will completely remake these figures (see General Response). The intended observation of **Fig. 4 (Pareto-frontier)** is the visualization of **Principle 3**:
> * The "best" configurations (top-left) are dominated by low-complexity methods.
> * Adding high-complexity techniques (moving right on the x-axis) rarely provides a proportional gain in score (y-axis).
> We will make this interpretation explicit in the revised caption and text.
>
> **Weakness 5: "Little insight; no deeper analysis."**
> We believe our three Principles constitute deep, actionable insights.
> * For example, **Principle 2** identifies that **Scale Representation** (not just gradient estimation) is the primary bottleneck. This explains *why* E4M3 fails on LLMs (insufficient range) and why E8M0 is suboptimal (insufficient precision).
> * This analysis directly led us to propose and validate **UE5M3** (Appendix C.3) as a superior alternative. We will move the UE5M3 analysis to the main paper to better showcase this insight.
>
> **Q1: "Definition of $M_{ref}$, $M_c$, penalty."**
> We apologize these definitions were located in the Appendix. We will move them to Section 7 in the revision:
> * $M_{ref}$: The validation loss of the **BFLOAT16** baseline.
> * $M_c$: The validation loss of the specific **FP4 configuration** $c$.
> * $\Omega(c)$: The complexity penalty defined in **Appendix B.1, Table 7**, calculated based on the overheads listed in Table 1.
>
> **Q2: "If E4M3 fails, would E8M0 work better?"**
> Yes, exactly. Our results show E8M0 works significantly better than E4M3 because E4M3 is limited by range.
> * **Llama 1B (Table 4):** "Pure FP4" E8M0 loss is **3.864**, whereas "Pure FP4" E4M3 is **6.815**.
> * However, our analysis suggests E8M0 lacks precision. This trade-off led us to test **UE5M3** (Appendix C.3), which offers a better range/precision balance and outperformed both formats.

---

### Official Review · Reviewer_QCDN · 2025-11-08

**Soundness:** 3
**Presentation:** 2
**Contribution:** 2
**Rating:** 2
**Confidence:** 3

**Summary:**

This paper presents a unified quantization-gradient framework for analyzing and stabilizing FP4 training in low-precision deep learning. Through extensive simulations, it identifies optimal configurations—combining Hadamard transforms, tensor scaling, and stochastic rounding—that achieve the best balance between performance and computational cost.

**Strengths:**

The paper goes through details of the math, existing corresponding methods, and examine on various tasks.

**Weaknesses:**

The novelty is limited and difficult to idenfity. The author claimed E5M3 outperforms than E8M0, but cannot find explicit comparison on many body or appendix. They found good consistency performance of some technique, but does not lead to significant finding, compared with SOTA, e.g., does it outperform SOTA MXFP4/NVFP4 training significantly?

Another diff betwen NV and MX is the blocksize, which is missing to ablate.

**Questions:**

Although the claim E5M3 outperforms than E8M0 is true, 4 bits is known to apply advanced technique like SR, RHT, QAT, author could not clearly present results under these techniques clearly, in terms of they are loselss. and very difficult to find, i.e., poor presentation. so which 4 bit recipe is the best, and second and how much of gain in terms of accuracy and speed? within standard deviation or not?

---

> ### Author Response · Authors · 2025-11-20
> **Response**
>
> ### Response to Specific Weaknesses
>
> **1. Response to: "Novelty is limited"**
> We respectfully clarify that our novelty lies in the **systematic unification and mapping** of the FP4 design space, rather than proposing a single heuristic. While prior work proposed isolated solutions, we contribute:
> * **A Common Framework (Sec. 2):** The first formal unification allowing analysis of disparate techniques.
> * **An Empirical Map (Sec. 7):** Determining what works and what doesn't across >20,000 configurations.
> * **Novel Insights:** We prove that complex gradient approximations are unnecessary (Principle 1) and identify scale format as the critical bottleneck (Principle 2)—findings that fundamentally shift the direction of future research.
>
> **2. Response to: "Cannot find explicit comparison for E5M3 vs E8M0"**
> We apologize that this key comparison was buried in the Appendix. We will move **Table 9 and Figure 9** (Appendix C.3) to the main paper to highlight this result.
> * **Evidence:** As shown in Table 9, our best Llama 1B **UE5M3** configuration reaches a validation loss of **3.586**.
> * **Comparison:** This outperforms the best **E8M0** (3.608, Table 4) and is far superior to the best **E4M3** (4.933, Table 4), providing strong empirical support for Principle 2.
>
> **3. Response to: "Does not lead to significant finding compared with SOTA"**
> Our goal was to scientifically elucidate *why* SOTA works, distinguishing necessary components from superstitious ones.
> * **Validation:** Our Principle 3 validates that specific SOTA techniques (SR, RHT) are mathematically and empirically superior.
> * **Pareto Dominance:** Figure 4 provides the evidence that these methods dominate the performance-overhead frontier, while other popular proposals (like gradient adjustments) do not.
> * **Revision:** We will add a "Best Recipe" summary table to explicitly contrast our recommended, efficient configuration against more complex, less effective SOTA baselines.
>
> **4. Response to: "Missing ablation on block size (NV vs MX)"**
> We utilized fixed block sizes (16 for E4M3, 32 for E8M0) to align with **standard industry practice** (e.g., Chmiel et al., 2025) and current **hardware constraints** (e.g., Nvidia/MX specifications require these specific fixed sizes). However, we agree that investigating block size sensitivity is valuable for future hardware definitions. We will add a discussion on these hardware constraints and list block size ablation as a key direction for future work.

---

### Author Response · Authors · 2025-11-20
**General Response to All Reviewers**

We thank all reviewers for their detailed feedback and time. We appreciate the insightful comments, which have highlighted several areas for improvement, particularly regarding presentation and clarity.

### 1. Clarifying the Core Contribution: Elucidating the Design Space
The central goal of this paper is to **"Elucidate the Design Space of FP4 Training."** The current field is fragmented; many recent papers propose isolated techniques (e.g., RHT, SR, new optimizers) without systematic comparison.

Our primary contribution is **not** to propose a single new SOTA technique, but rather to provide:
* **A Unified Gradient-Based Framework (Sec. 2):** We provide the first formal analysis and overhead comparison of disparate stabilization methods (e.g., differentiable absmax, tensor scaling gradients), grounded in Proposition 1 and Corollary 1.
* **A Large-Scale Empirical Study (Sec. 7):** We systematically evaluated **>20,000 configurations** to map the interactions and performance-vs-overhead trade-offs across diverse tasks (Regression, Classification, Diffusion, LLMs).

### 2. Significance of Key Findings
Our results bring essential clarity to a complex space. We highlight three principles that directly impact future research and hardware design:

* **Principle 1: Simplicity prevails (STE > Complex Approximations).**
    This is a crucial, non-obvious result. We demonstrate that computationally expensive gradient approximations (including our own proposed splines and differentiable absmax) **do not outperform** the simple, efficient Straight-Through Estimator (STE). This contradicts assumptions in recent works (e.g., *Wang et al., 2025*) and justifies a significantly simpler training pipeline.

* **Principle 2: Scale format is the primary bottleneck.**
    Prior work often overlooked the specific impact of scale formats. We introduce **UE5M3** to this analysis (Appendix C.3) and demonstrate that it provides a superior range/precision compromise, outperforming both **E8M0 (MXFP4)** and **E4M3 (NVFP4)** on LLMs. This is a vital insight for future hardware/software co-design.

* **Principle 3: The performance frontier is sparse.**
    We confirm *why* SOTA recipes work: the Pareto frontier is dominated by a few low-overhead techniques (**RHT/Hadamard, SR, Tensor Scaling**). Our Pareto plots (Fig. 4) are the first to quantify this, guiding practitioners to avoid high-complexity, low-return methods.

### 3. Action Plan: Addressing Presentation and Clarity
We acknowledge the universal feedback regarding figure clarity and presentation flow. This was an oversight, and we commit to a full revision for the camera-ready version to ensure the narrative matches the strength of the empirical data:
1.  **Visualization Overhaul:** We are completely remaking **Figures 1, 3, and 4** with larger fonts, clearer legends, and non-overlapping axes to ensure readability.
2.  **Highlighting Key Results:** We will move the **UE5M3 LLM comparisons** (currently in Appendix C.3) into the main paper to directly support Principle 2.
3.  **Explicit Narrative:** We will revise the text to explicitly link the theoretical framework and experiments to the three Principles outlined above.
4. Contrast against the recent Nvidia paper https://arxiv.org/pdf/2509.25149

We believe these revisions will resolve the primary concerns and clearly demonstrate the paper's significant contribution to the FP4 training landscape.

---

### Author Response · Authors · 2025-11-27
**General comment after discussions**

We thank the reviewers for the their time and valuable comments. We acknowledge the validity of the points raised in the discussions with them, and as it requires extensive large scale experiments to answer them with confidence, we will not be able to get the results in time for the rebuttal deadline. But we are committed to improve and cover these points in further iterations of our work.

---

### Meta-Review · Area_Chair_JFhD · 2025-12-31

**Summary:**

This paper explores FP4 training of deep neural networks using the MxFP4 and NvFP4 data formats. The paper experiments with various combinations of techniques and hyperparameters commonly used in FP4 training and recommends configurations that provide a reasonable balance between performance and overhead.

**Reviewer Concerns:**

1. Reference accuracy issues: Several claims about prior work are inaccurate/misleading
2. No solid empirical studies include, nor a deep analysis.
3. Writing/presentation quality: Poor figure readability (tiny legends, labels, line width, overlapping axes figure 4d), inconsistent paragraph formatting (spacing, bold text), unclear symbols meaning

**Reviewer Scores:**

The paper received all negative ratings initially. The authors provided a rebuttal to address some of the concerns of the reviewers. However, the main concerns are not addressed. AC thought the paper is more like a technical report instead of a research, and is below the bar of ICLR. The final decision is rejection.

---

### Decision · Program_Chairs · 2026-01-26

Reject